# UNREALZOO: ENRICHING PHOTO-REALISTIC VIRTUAL WORLDS FOR EMBODIED AI AGENTS

## ABSTRACT

The embodied artificial intelligence agents should be capable of sensing, reasoning, planning, and acting in complex open worlds, which are unstructured, high-dynamic, and uncertain. To apply agents in the real world, the realism of the simulated worlds is important for training and evaluating the built agents. This paper introduces UnrealZoo [1], a rich collection of photo-realistic 3D environments that mimic the complexity and variability of the real world based on Unreal Engine. For embodied AI, we provide a diverse array of playable entities in the environments and a suite of tools, based on UnrealCV, for data collection, reinforcement learning, and evaluation. In the experiments, we benchmark the agent on visual navigation and tracking, two fundamental tasks for embodied vision agents, in complex open worlds. The results provide valuable insights into the strengths of enriching the diversity of the training environments and the challenges to current embodied vision agents in the open worlds, e.g., the latency in the closed-loop control to interact with the dynamic objects, reasoning the accordance of the spatial structure in the complex scenes.

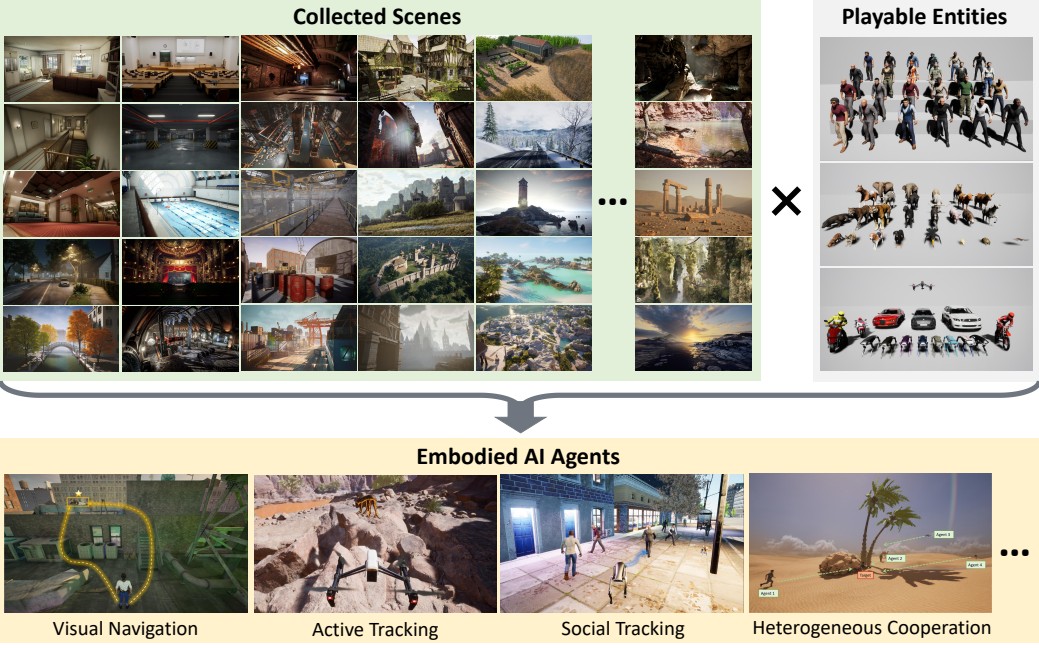

Figure 1: UnrealZoo enriches photo-realistic virtual worlds for embodied AI research by aggregating diverse scenes and playable entities. These environments facilitate the training and testing of embodied AI agents on tasks such as visual navigation, social tracking, and multi-agent cooperation, addressing challenges in open-world deployments.

---

[1]Project page: `https://unrealzoo.notion.site/`

# 1 INTRODUCTION

Currently, embodied artificial intelligence (Embodied AI) agents are often *homebodies*, primarily confined to controlled indoor environments and rarely venturing outside to explore the diversity of the open world. While several simulators have advanced the field, including AI2-Thor (Kolve et al., 2017), OmniGibson (Li et al., 2023), VirtualHome (Puig et al., 2018), and Habitat (Puig et al., 2024), they often focus on specific scenarios, such as daily activities in homes, which limits the development of generalist embodied AI in open worlds. The lack of richness and variability in simulators hampers agents' ability to adapt and generalize to the diverse challenges of real-world environments, from bustling urban areas to rugged natural landscapes.

Thus, it is crucial to build diverse photo-realistic 3D virtual environments to simulate challenges in open worlds for advancing embodied AI. Such environments will help agents develop robust skills to sense, reason, plan, and control for accomplishing various tasks. By simulating complex scenarios and interactions, researchers can evaluate how embodied agents respond to uncertainty, adapt to dynamic challenges, and learn from their experiences in a controlled yet rich context. This process not only fosters the development of sophisticated perception and decision-making abilities but also enhances the agents' capacity to collaborate with humans and other AI systems, paving the way for seamless integration into real-world applications. As these virtual worlds grow increasingly sophisticated, incorporating realistic physics, intricate social dynamics, and varying levels of abstraction, they offer the potential for agents to experience a wider spectrum of situations. This diversity is essential for training robust agents that can generalize well to unseen environments and tasks. Furthermore, the iterative feedback loop between the agents and environments will enable continuous improvement, allowing agents to refine their skills through both simulated challenges and real-world encounters.

In this work, we introduce UnrealZoo, a comprehensive collection of photo-realistic virtual environments set, based on Unreal Engine [2] and UnrealCV (Qiu et al., 2017), featuring a diverse range of complex open worlds and playable entities to advance research in embodied AI. This high-quality set encompasses a wide range of complex indoor and outdoor scenes, such as houses, supermarkets, train stations, industrial factories, villages, temples, and natural landscapes, providing a platform to study how AI agents perceive and interact within a variety of complex dynamic environments. Each environment is carefully crafted by artists to replicate realistic lighting, textures, and dynamics, closely resembling real-world experiences. Our collection also includes diverse entities—humans, animals, robots, drones, motorbikes, and cars—each with unique appearances and movements, enabling researchers to investigate the generalization of the agents on different embodiments. To enhance usability, we have optimized UnrealCV and offer a suite of tools and APIs (UnrealCV+), including environment augmentation, demonstration collection, and distributed training/testing. These tools allow customization and extension of the environments to meet various research needs. This flexibility ensures UnrealZoo remains adaptable as the field of embodied AI evolves.

We conduct experiments to demonstrate the applicability of UnrealZoo for embodied AI. First, we benchmark frames per second (FPS) across various commands, highlighting the significant improvement in image rendering and multi-agent interactions with the UnrealCV+ API. We use embodied visual navigation and tracking as two example tasks to benchmark embodied vision agents in complex dynamic environments with moving objects and unstructured maps. We also introduce a set of simple yet effective baseline methods for developing embodied vision agents, including distributed online reinforcement learning algorithms, offline reinforcement learning algorithms, and a reasoning framework for large vision-language models (VLMs). Our evaluations across different settings emphasize the importance of diverse training environments for enhancing agent generalization and robustness, the necessity of low latency in closed-loop control to handle dynamic factors, and the potential of reinforcement learning for training agents to navigate complex scenes.

Our contributions can be summarized in the following: 1) We build UnrealZoo, a collection of 100 high-quality photo-realistic scenes and a set of playable entities with diverse features, covering the most challenging to embodied AI agents in open worlds. 2) We optimize the communication efficiency of UnrealCV APIs and provide easy-to-use Gym interfaces with a toolkit for diverse requirements. 3) We conduct experiments to demonstrate the usability of UnrealZoo, showing the importance of the diversity of the environments to the embodied agents, and analyzing the limitations of the current RL-based and VLM-based agents in the open worlds.

---

[2]www.unrealengine.com

Table 1: The comparison with related photo-realistic virtual worlds for embodied AI. The comparison of visual realism across different engines is presented in Figure 6 and the emoji descriptions are listed in Table 6. (**Unstr. Terr.** indicating the presence of unstructured terrain. **Nav. Sys.** specifying whether the agent in the environment includes an autonomous navigation system.)

| Virtual Worlds | Scene: Categories | Scene: Scale Level | Scene: Unstr. Terr. | Scene: Base Engine | Agent: Body | Agent: Nav. Sys. | Agent: Multi-agent |
|---|---|---|---|---|---|---|---|
| VirtualHome | 🏠 | Room | - | Unity | | ✓ | ✓ |
| AI2THOR | 🏠 | Room | - | Unity | 🧑‍🦽🤖 | - | - |
| ThreeDWorld | 🏠🏯⛰ | Room, Building, Landscape | ✓ | Unity | 🧑‍🦽🤖✈🚗 | - | ✓ |
| OmniGibson | 🏠 | Room | - | Omniverse | - | - | - |
| Habitat 3.0 | 🏠 | Room | - | Habitat-Sim | 🧑‍🦽🤖 | ✓ | ✓ |
| CARLA | 🏢 | Building, Town | - | UE 4 | 🚗 | - | ✓ |
| AirSim | 🏢 | Building, Town, Landscape | - | UE 4 | 🚗✈ | - | ✓ |
| LEGENT | 🏠🏢 | Room, Building | ✓ | Unity | 🧑‍🦽🤖 | ✓ | - |
| V-IRL | 🏢⛰🏰 | Town, Landscape | ✓ | Google Map | 📷 | ✓ | ✓ |
| **UnrealZoo** | 🏠🏯⛰🏰🏯🏢🏙🏖 | Room, Building, Town, Landscape | ✓ | UE 4/5 | 🧑‍🦽🤖🚗🐫🧍🏍✈📷 | ✓ | ✓ |

## 2 RELATED WORKS

**Realistic Simulators for Embodied AI.** Realistic simulators are extensively utilized in embodied artificial intelligence due to their appealing benefits, including high-quality rendering, cost-effective ground truth generation, low-cost interaction, and environmental controllability. They are crucial for training and testing AI agents to handle increasingly complex tasks. Notable realistic 3D simulators have been created for specific applications, such as indoor navigation (Kolve et al., 2017; Puig et al., 2018; Xia et al., 2018; Wu et al., 2018), robot manipulation (Yu et al., 2020; Ehsani et al., 2021; Chen et al., 2024), and autonomous driving (Gaidon et al., 2016; Shah et al., 2018; Dosovitskiy et al., 2017). Recent advances in computer graphics have spurred interest in developing general-purpose virtual worlds with photo-realistic rendering, allowing agents to collect high-fidelity data and learn skills applicable across various tasks and scenes. ThreeDWorlds (TDW) (Gan et al., 2021) and LEGENT (Cheng et al., 2024) are notable simulators that offer photo-realistic, multi-modal platforms, based on Unity, for interactive physical simulation. However, their built-in scenes and playable entities are somewhat limited. Additionally, the performance of the simulator decreases significantly in large outdoor environments, a typical weakness of Unity. V-IRL (Yang et al., 2024) is a recent approach that leverages Google Maps' API to simulate agents with real-world street view images, significantly reducing the gap between virtual and real-world settings. However, since V-IRL is inherently composed of static images, it lacks the capability to simulate the dynamics of the physical worlds for agent-object interactions. Recently, the community has also begun to explore dynamic environments with social interactions and unexpected events. However, existing solutions like Habitat 3.0 (Puig et al., 2024) focus on a limited number of agent interactions in indoor scenes, while HAZARD (Zhou et al., 2024b) addresses only single-agent simulations in dynamic scenarios like fires, floods, and winds. In contrast, UnrealZoo offers a comprehensive collection of scenes that feature dynamic situations and diverse playable entities for embodied AI. With advancements in Unreal Engine and optimized UnrealCV, our environment achieves real-time performance in large-scale scenes with multiple agents (around 10) and photo-realistic rendering. A comprehensive comparison across the related photo-realistic simulators is shown in Table 1.

**Embodied Vision Agents.** Embodied vision agents, which perceive and interact with their environments through vision, are a key focus in artificial intelligence research. These agents perform tasks like navigation (Zhu et al., 2017; Gupta et al., 2017; Yokoyama et al., 2024; Long et al., 2024), active object tracking (Luo et al., 2018; Zhong et al., 2019; 2021; 2023; 2024), and other interactive tasks (Chaplot et al., 2020; Weihs et al., 2021; Ci et al., 2023; Wang et al., 2023), mimicking human behavior. Their development involves various methods, including state representation learning (Yadav et al., 2023; Yuan et al., 2022; Gadre et al., 2022; Yang et al., 2023), reinforcement learning (RL) (Schulman et al., 2017; Xu et al., 2024; Ma et al., 2024), and large vision-language models (VLMs) (Zhang et al., 2024; Zhou et al., 2024a). Despite significant progress, challenges remain. RL methods often require extensive trial-and-error interactions and computational resources for training, and they usually struggle to generalize to new environments. Conversely, VLM-based methods excel at interpreting language instructions and images but may lack the fine-grained control and adaptability necessary for real-time interactions. The computational demands and time needed for inference with such large models are critical, especially in dynamic scenes. Moreover, previous simulators mainly

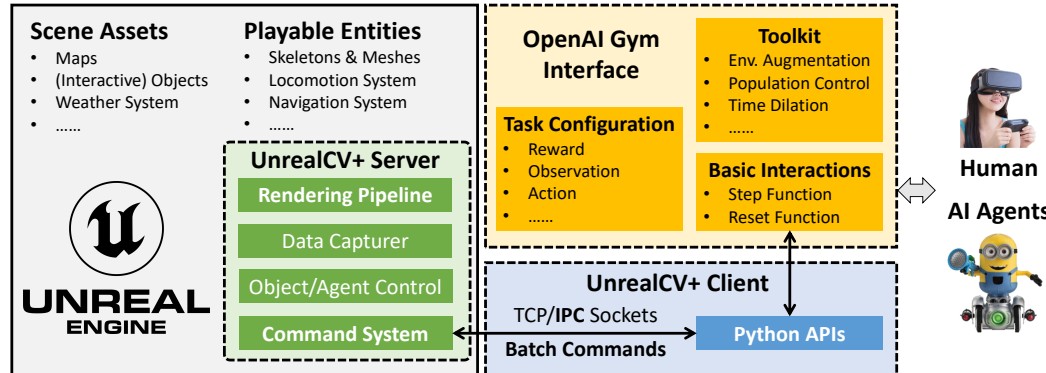

Figure 2: The detailed architecture of UnrealZoo. The Gray box indicates the UE binary, collecting the scenes and playable entities. The UnrealCV+ Server is built in the binary as a plugin. We have bolded the names of the optimized or new modules in UnrealCV+ Server and Client. For agent-environment interaction, we provide OpenAI Gym Interface, which has been widely used in the community. Our gym interface supports customizing the task in a configuration file and contains a toolkit with a set of gym wrappers for environment augmentation, population control, etc.

focus on indoor rooms or urban roads, which mask the potential challenges to the embodied agents when deploying in open worlds, e.g., unstructured terrain, dynamic changing factors, inference costs of the perception-control loop, and social interactions with other agents. Therefore, it is required to benchmark agents in large-scale, photo-realistic open worlds, taking into account various real-world challenges in the virtual worlds. In this work, we collect a subset of environments from UnrealZoo and benchmark embodied visual navigation and tracking agents, to emphasize the weakness of the existing methods.

## 3 UNREALZOO

UnrealZoo is a collection of photo-realistic, interactive open-world environments with diverse embodied characters, built on Unreal Engine and UnrealCV (Qiu et al., 2017). The environments are sourced from the *Unreal Engine Marketplace* [3], which shares 3D resources from artists, and were accumulated over two years at a cost exceeding $10,000$. UnrealZoo features a diverse array of scenes with varying sizes and styles. Among them, the largest scene, i.e., Medieval Nature Environment, covers more than $16km^2$ areas. The environments also include a wide range of embodiment, such as human avatars, vehicles, drones, animals, and virtual cameras, all of which can interact with the environment and equipped with ego-centric sensing systems. We offer easy-to-use Python APIs based on UnrealCV to facilitate interaction between Python programs and the game engine. Note that UnrealCV is optimized for rendering and communication, particularly in large-scale and multi-agent scenarios, namely UnrealCV+. Additionally, we provide OpenAI Gym interfaces to standardize agent-environment interactions. The gym-like interface also contains a set of toolkits, e.g., environment augmentation, population control, time dilation, and JSON-style task configurations to help the user customize the environments for various tasks with minimal effort. The project website includes the details of the collected contents, and documents about tutorials, Python APIs, and the gym interface.

### 3.1 SCENE COLLECTION

UnrealCV Zoo contains 100 scenes based on Unreal Engine 4 and 5. We select the scene based on the public reviews in the marketplace and the difference to the collected scenes, aiming at covering a wide range of styles from realistic to fictional, ensuring diversity. We provide an overview of the environments in the scene gallery.

---

[3]https://www.unrealengine.com/marketplace

We have tagged the collected scenes with a number of feature labels allowing researchers to select appropriate scenes for testing or training based on the tags associated with each scene. Our tags cover the following aspects:

- **Scene Categories**: We categorize scenes into three main types: interior, exterior, and both. The interiors include private houses, museums, supermarkets, train stations, factories, gyms, and caves. The exteriors include various outdoor terrains such as ruins, islands, plazas, neighborhoods, and mountains. Additionally, there are 25 scenes that feature both interior and exterior features, requiring enhanced spatial reasoning to comprehend their structure.

- **Spatial Structure**: We also tag the spatial structure of the scenes, including multi-floor, topological, flat, steep, etc. Such categorization is vital for benchmarking embodied agents, where the agent's performance is greatly impacted by the geometric structures.

- **Dynamics**: Environments featuring significant weather and animation change that create random dynamic lighting variations, visual obstructions, and effects such as sandstorms, snowfall, and thunderstorms, enhancing the environment's interference with visual tasks. Besides, we also label the environments with *interactive objects* where agents can interact with objects, e.g., open a door. These dynamics are essential for the open world.

- **Scale**: Each scene is labeled by the scale, such as small (room-level), medium (building-level), and large (city-level or landscape-level). Extra-large maps with delicate buildings, terrain, and illumination, are available for complex, long-term tasks such as rescues, which require extensive environments and high exploration needs.

- **Style**: The scenes may also reflect different cultural backgrounds, such as *Asian Temple*, *Western Church*, *Middle East Street*, or *Modern City*, *Science Fiction*. Identifying cultural styles will help us build a new data set to benchmark how social agents adapt to diverse cultures and social norms.

After categorizing the scenes, we integrate UnrealCV+ into the UE project (Refer to Section 3.3) and add the controllable player assets (Refer to Section 3.2) to each scene. Due to licensing restrictions, content purchased from the marketplace cannot be open-source, so we will package the projects into an executable binary for sharing with the community. These executable binaries will be compatible with various operating systems, including Windows and Linux, allowing users to download and run them via the Python interface without needing any knowledge of Unreal Engine, which is primarily built on C++ and Blueprint.

## 3.2 PLAYABLE ENTITIES

UnrealZoo includes seven types of entities: humans, animals, cars, motorbikes, drones, mobile robots, and flying cameras (See Figure 1). Specifically, it comprises 19 human entities, 27 animal entities, 3 cars, 14 quadruped robots, 3 motorcycles, and 1 quadcopter drone. This diversity, with varying affordances like action space and viewpoint, allows us to explore new challenges in embodied AI, such as cross-embodiment generalization and heterogeneous multi-agent interactions.

Each entity includes a skeleton with appropriate meshes and textures, a local motion system, and a navigation system. We offer a set of callable functions for each entity, enabling users to modify attributes like size, appearance, and camera positions, as well as control movements. Each entity can switch between different textures and appearances via UnrealCV API, enhancing visual diversity

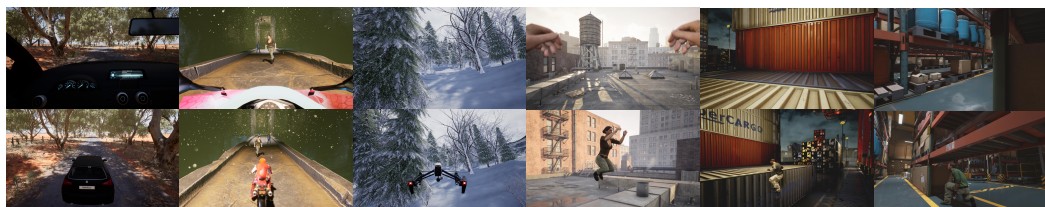

Figure 3: The first-person view images (Top) while playing different entities and motions across various scenes.

Table 2: Comparison of FPS in Unreal Engine 4.27 with UnrealCV and UnrealCV+.

| | Image Capture | | | | Multi-agent Interaction | | |
|---|---|---|---|---|---|---|---|
| | Color | Obj. Mask | Suf. | NorDepth | N=2 | N=6 | N=10 |
| UnrealCV | 74 | 70 | 109 | 52 | 35 | 13 | 8 |
| UnrealCV+ | 83(↑ 12%) | 154(↑ 120%) | 131(↑ 20%) | 97(↑ 86%) | 54(↑ 54%) | 25(↑ 92%) | 16(↑ 100%) |

and adaptability for various scenarios. Each entity is equipped with an ego-centric camera, allowing researchers to capture various types of image data such as RGB, depth, surface normal, and instance-level segmentation from the agent's ego-centric view. Figure 3 shows examples of the captured first-person view and third-person view images of different entities with varying locomotion. For multi-agent interaction, the population of the entities in a scene can be easily adjusted using the spawn or destroy functions.

**The locomotion system** is built on Smart Locomotion, a well-designed and smooth locomotion system. It contains a number of high-quality animations that enable the agent to interact with the scene, such as opening and closing doors, crouching under obstacles, jumping over obstacles, climbing onto a platform, and simulating injury or death. With the locomotion system, we can explore the agent's ability to reason, plan, and interact in large-scale complex 3D scenes in advance, ignoring learning skills for low-level action control that requires high-fidelity physical simulation.

**The navigation system** is built on NavMesh allowing agents to autonomously navigate with the built-in AI controller in the Unreal Engine. This includes path-finding and obstacle-avoidance capabilities, ensuring smooth and realistic movement throughout diverse terrains and structures. For urban-style maps, we segment the roads to distinguish between pedestrian and vehicle pathways. When agents use the navigation system for autonomous control, they will navigate the shortest path based on the priority of the different areas. For example, pedestrians and animals will prioritize walking on sidewalks, while vehicles and motorcycles will prioritize driving on roadways. An example of the navigation area is shown in Figure 9.

### 3.3 PROGRAMMING INTERFACE

We provide UnrealCV+ as the basic application programming interface (API) on Python to capture data and control the entities and scenes, and provide an OpenAI Gym interface for general agent-environment interactions. The architectures of the programming interfaces are shown in Figure 2.

**UnrealCV+** is our modification version of the UnrealCV (Qiu et al., 2017) for high-throughput interactions. As the original version of UrnealCV primarily focuses on data generation, the frame rates per second (FPS) are not optimized for real-time interactions. We optimize the rendering pipelines in UnrealCV Server and the communication protocols between UnrealCV Server and Client to improve the FPS. Specifically, we enable parallel processing while rendering object masks and depth images, which can significantly improve the FPS in large-scale scenes. For multi-agent interactions, we further introduce the batch commands protocol. In this protocol, the client can simultaneously send a batch of commands to the server, which can process all the received commands and return a batch of results. In this way, we can reduce the time spent on server-client communication. Since reinforcement learning requires an extensive number of trial-and-error interactions for training, often running multiple environments on a computer, we additionally introduce Inter-process communication (IPC) sockets instead of the TCP sockets to improve the stability of the server-client communication under high loads. We benchmark the FPS performance in Table 2. To enhance user-friendliness, we have developed high-level Python APIs that are built upon the command systems of UnrealCV. These APIs encapsulate all the request commands and their corresponding data decoders into a callable Python function. This approach significantly simplifies the process for beginners, allowing them to interact with and customize the environment using UnrealCV+.

**OpenAI Gym Interface** is used to define the tasks and standardize the agent-environment interaction, following Gym-UnrealCV. Even though there are a lot of tasks for agents, they usually share common interaction protocols, i.e., the agent gets observations from the environment and returns actions. The main difference across different tasks usually is the reward functions, the modality of the observation, and the available actions. Hence, we define the basic interaction functions for general usage and list the task-specific configurations, e.g., scene name, and reward function, in a JSON File, as shown in Figure 11. In this way, when adding new UE scenes, the users only need to set the parameters in the

JSON files. Moreover, we contain a toolkit with a set of gym wrappers for training and testing the agents, such as environment augmentation that has been in previous work for training generalizable agents (Luo et al., 2018; 2020), population control to adjust the number of agents in the scene, and time dilation to adjust the control frequency in dynamic scenes. In Section 4.3, we demonstrate an example usage of the toolkit to analyze the robustness of social tracking agents to the population of crowds and the impact of the control frequency in such dynamic scenes. We also provide a launch tool to enable the user to run multiple environments with specific GPU IDs within a computer, which is useful for distributed online reinforcement learning.

## 4 EXPERIMENTS

In this section, we use a subset of UnrealZoo to demonstrate the usability of the collected environments. For visual navigation, we select two scenes with complex spatial structures to train and validate the RL-based and VLM-based agents. For active tracking, we select at most 8 scenes as training environments and validate the generalization of the learned policy in another 24 scenes, which are divided into four categories according to the scene types. The results demonstrate the importance of the diversity of the training environments to the cross-domain generalization. For social tracking, we analyze the robustness of the agent in social environments with different control frequencies, using the toolkit provided in the gym to generate crowds with varying populations and control frequencies.

### 4.1 VISUAL NAVIGATION

In-the-wild visual navigation introduces a new level of complexity compared to traditional navigation tasks for indoor scenes or autonomous driving, which often run on structured maps. Differently, we place the agent in open-world environments where it must take a set of locomotions, e.g., running, climbing, jumping, crouching, to go over the various obstacles in unstructured terrains to reach the target object. In this setting, the agent requires advanced scene reasoning and action affordance to make real-time decisions about its path. The emphasis on such complex environments ensures the agent can operate effectively in a broad range of challenging scenarios, moving beyond the constraints of traditional navigation frameworks. The details of the task setting are introduced in Appendix B.1.

**Evaluation Metrics.** We employ two key metrics to evaluate visual navigation agents: 1) Average Episode Length (EL), representing the average number of steps per episode over 50 episodes. 2) Success Rate (SR), measuring the percentage of episodes the agent successfully navigates to the target object out of 50 total episodes, which represents the navigation capability in the wild environment.

**Baselines for Navigation.** We build simple baselines to demonstrate the applicability of our environments for training reinforcement learning agents and benchmark the agents based on pre-trained large models. **1) Online RL**: We trained the RL-based navigation agents separately in the Roof and Factory environments using a distributed online reinforcement learning (RL) approach, e.g. A3C (Mnih et al., 2016). The training curve is shown in Figure 15. The model takes the first-person view segmentation mask and the relative position between the agent and target as input, and outputs direct control signals (from the predefined action space) to navigate. This setup allows the agent to learn and optimize navigation strategies during continuous interaction with the environment. Please refer to Appendix C.1 for the implementation details. **2) GPT-4o**: We employ the GPT-4o model to take action, leveraging its powerful multi-modal reasoning capabilities. The model takes first-person view images and the relative position between the agent and the fixed target as input. The GPT-4o model follows our prompt template (See Table 13) as guidance, reasoning appropriate actions from the predefined control space to guide the agent toward the target. **3) Human**: We also have a human player control the agent using a keyboard, similar to a first-person video game. The player navigates the agent from a random starting point to a fixed target, making decisions based on visual observations from the shared control space.

**Results.** In Table 3, we report the performances of different methods in two unstructured scenes. The RL-based agent performs moderately well, achieving better results in the simpler IndustrialArea environment compared to the Roof environment, where the target object is located on different levels of stairs. The agent based on GPT-4o struggles in both scenarios. This infers that the GPT-4o performs poorly in complex 3D scene reasoning. As a reference, the human player completes both tasks with the fewest steps and a 1.00 success rate, underscoring the significant gap between current embodied AI agents and human performance, indicating substantial room for improvement to navigate in such complex, open-world environments.

Figure 4: An exemplar sequence from the RL-based agent in the *Roof*.

## 4.2 ACTIVE VISUAL TRACKING

We evaluate the generalization of the tracking agents across four environment categories: **Interior Scenes**, **Palaces**, **Wilds**, and **Modern Scenes**. Each category contains 4 individual environments, as shown in Figure 8. We aim to capture a broad range of features in our environment collection by selecting four distinct and representative scenes from each category, ensuring a comprehensive eval-

Table 3: The results (EL/SR) of in-the-wild visual navigation in two unstructured terrains.

| Methods | Roof | IndustrialArea |
|---|---|---|
| Online RL | 1660/0.32 | 261/0.52 |
| GPT-4o | 2000/0.00 | 369/0.20 |
| Human | 515/1.00 | 158/1.00 |

uation of the agents' capabilities. The details of the tasks are introduced in Appendix B.2. We analyzed the effectiveness of the diversity of the training data by collecting demonstrations with different numbers of training environments.

**Evaluation Metrics.** Our evaluation employs three key metrics: (1) Average Episodic Return (ER), which calculates the mean episodic return over 50 episodes, providing insights into overall tracking performance; (2) Average Episode Length (EL), representing the average number of steps per episode, which reflects long-term tracking effectiveness; and (3) Success Rate (SR), measuring the percentage of episodes that complete 500 steps out of 50 total episodes.

**Baselines for Active Tracking.** For the *RL-based agents*, we extend from the official implementation settings from the recent offline RL method (Zhong et al., 2024), collecting offline datasets and employing the original network architecture. To demonstrate the impact of data diversity on tracking performance, we collect three sets of offline datasets, each containing 100k steps. The key difference between these datasets is the number of environments used for data collection: one was collected in a single environment (denoted as *1 Env.*), another in two environments(denoted as *2 Envs.*), and the third in eight distinct environments (denoted as *8 Envs.*). The offline training curve of each setting is shown in Figure 14. The environment distribution of each dataset setting is shown in Figure 10. It is worth noting that FlexibleRoom, one of the environments used for data collection, is a unique abstract environment, with all objects represented as geometric shapes covered by randomized patterns. This distinctive setup contrasts with the more realistic and diverse environments in the collection, offering a unique scenario for testing agent adaptability. For the *VLM-based agents*, we utilize the latest large models GPT-4o to directly generate actions based on observed images for tracking a target person. To ensure smooth and precise transitions, we designed a system prompt that helps the model understand the task while standardizing the output format to align with predefined action settings. This prompt ensures the model produces actions coherent with the task's requirements. Specifically, GPT-4o is tasked with generating concrete action decisions from a predefined instruction space: moving forward, moving backward, turning left, turning right, or maintaining the current position. Once an instruction is generated, we map it to corresponding linear and angular velocities to update the agent's movement in the environment. It is important to note that while the system prompt can use raw image observations as input, our experience shows poor alignment performance and significant time delays, which pose challenges for real-time tracking. The full system prompt and mapping relationship are provided in Appendix C.2.

**Result Analysis.** We first evaluate the performance of agents trained with offline datasets collected from varying numbers of environments (1 Env., 2 Envs., 8 Envs.) across **16 distinct environments**. We list the detailed evaluation results across the entire 16 environments in Table 11. To better visualize

the performance change of different training settings within various scene categories, we calculate the average success rate (SR) of each agent in four categories, the results are shown in Figure 5. The results reveal a clear trend: **as the number of environments used for training increases, agent long-term tracking performance generally improves across all categories.** In the Wilds, a significant increase in success rate is observed with the 8 Envs. dataset, which involves the highest diversity of environments. This demonstrates that diverse environmental exposure plays a crucial role in improving the agent's generalization capabilities in more complex, open-world environments. The lower success rate in the 1 Env. dataset highlights the limitations of training solely in abstract settings like the FlexibleRoom. Similarly, in the Palace, the success rate improves notably from 1 Env. to 8 Envs., suggesting that training with a broader range of environments helps the agent better adapt to intricate spatial structures typical of Palace-like maze environments.

## 4.3 SOCIAL TRACKING

We further evaluate the tracking agents in a social tracking setting, where the agent needs to follow the target in crowds. Such a setting contains varying high-dynamics objects with similar appearances. We can directly apply the toolkit provided in the Gym interface to extend the *DowntownWest* environment used for active tracking to this setting.

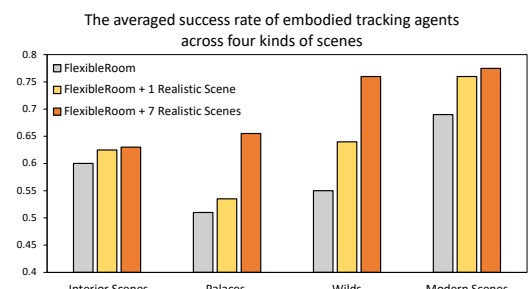

**Robustness to Active Distractions.** A key challenge in active visual tracking tasks is managing active distractions, a critical issue for real-world deployment in crowds. Thus, we conducted an experiment in the DowntownWest environment and generated crowds with varying numbers of human characters as distractors notated as 4D, 8D, and 10D. We compared the performance of the offline RL method, trained under three dataset configurations (1 Env., 2 Envs., 8 Envs.), against the VLM-based method, evaluating the agents' ability to maintain robust tracking under these different levels of active distractions. The results in Table 5 show clear performance differences between the offline RL methods (1 Env., 2 Envs., 8 Envs.) and the GPT-4o model in

Figure 5: Average success rate of agents across four environment categories: Compact Interior, Wildscape Realm, Palace Maze, and Lifelike Urbanity, evaluated under three offline dataset settings (1 Env, 2 Envs., 8 Envs.). The results show the generalization capability improves significantly as more diverse environments are included in the dataset. However, environments with complex spatial structures, such as Compact Interior and Palace Maze, exhibit lower success rates, highlighting challenges in obstacle avoidance and navigation.

handling active distractions. As the number of distractors increases, the offline RL methods maintain relatively stable success rates (SR), with the highest performance seen in the 8 Envs. setting, which achieves an SR of 0.8 in the 4D condition and remains robust with slight declines in the 8D and 10D conditions (0.72 and 0.68, respectively). This suggests that the agent benefits from the richer diversity of training data, enabling it to handle increasingly complex crowd scenarios more effectively. On the other hand, the GPT-4o model consistently struggles with active distractions, showing significantly lower average returns (ER) and success rates across all settings. The model's inability to cope with dynamic, crowded environments is evidenced by its poor performance, particularly in the 10D condition where it records a success rate of just 0.1. This highlights a major limitation of the VLM-based method in dynamic environments with active distractions, as it lacks the temporal consistency and real-time adaptabilit required for effective tracking.

**Cross-Embodiment Generalization.** We transfer the agent trained for the human character to the robot dog, which observes the world from a lower perspective. We can see that the results in Table 5 drop, particularly the success rate, indicating that the research community should pay more attention to the cross-embodiment generalization.

**The Impact of Control Frequency.** We employ the time dilation wrapper to simulate different control frequencies during deployment. The frequency of the perception-control loop is crucial for handling dynamic environments. As is shown in Table 4, when the rate drops below 10 FPS, performance significantly declines. We observe that higher control frequencies enable RL-based

Table 5: Performance comparison of different methods in the DowntownWest environment with varying numbers of distractors (4D, 8D, 10D). Each cell presents three metrics from left to right: Average Episodic Return (ER), Average Episode Length (EL), and Success Rate (SR).

| Method | 4D | 8D | 10D |
|---|---|---|---|
| Offline RL 1 Env. | 251/450/0.70 | 201/406/0.58 | 230/247/0.64 |
| Offline RL 2 Envs. | 309/456/0.74 | 259/424/0.68 | 258/428/0.68 |
| Offline RL 8 Envs. | 245/458/0.80 | 225/435/0.72 | 218/444/0.68 |
| Offline RL 8 Envs. (Robot dog) | 220/409/0.48 | 189/386/0.42 | 143/367/0.40 |
| GPT-4o | -102/264/0.16 | -64/270/0.14 | -80/240/0.10 |

agents to perform better in social tracking. These results emphasize the importance of building efficient models for embodied agents, to accomplish tasks in dynamic open worlds.

## 4.4 LIMITATION ANALYSIS AND SUMMARY

The current RL method shows some capacity to learn spatial-temporal information and dynamically respond to target movement in most scenarios, but it struggles with executing advanced actions like bypassing obstacles. In compact Interior categories and some special environments such as TerrainDemo, IndustrialArea, and ModularSciFiSeason1, which feature irregular landscapes, narrow passageways, and maze-like structures, the agent often collides with casually placed low-level objects. While the agent can track targets, its insufficient to handle unpredictable hindrances, especially in key moments like bypassing corners or tight spaces, which increases the likelihood of failure. This highlights a significant limitation: although the agent can learn and react to its environment, it lacks the higher-level reasoning to anticipate and avoid obstacles effectively. Advanced behaviors like bypassing obstacles are crucial for improving performance, especially in cluttered environments where basic reactive controls are insufficient. Incorporating such reasoning mechanisms would help reduce failure rates, particularly in critical scenarios, and improve overall tracking performance.

Table 4: The impact of control frequency on tracking performance. We evaluate the agent (Offline RL 1 Env.) in the FlexibleRoom environment using the time dilation wrapper to simulate varying control frequencies.

| | ER/ EL/ SR. |
|---|---|
| 3 FPS | 184/377/0.34 |
| 10 FPS | 303/449/0.62 |
| 30 FPS | 368/482/0.92 |
| w/o Control | 275/425/0.74 |

For the VLM-based method, one key factor contributing to GPT-4o's notably poor performance, especially in comparison to the RL methods, is its susceptibility to time delays. From our experience, this issue becomes particularly evident when the target makes abrupt movements, such as turning around. Due to the API's response lag, the GPT-4o system struggles to track the target in real-time, often losing it before receiving updated instructions. This limitation highlights the difficulty of real-time processing in embodied tracking tasks using models that rely on slower external API communications, underscoring the need for more efficient integration methods for such systems.

## 5 CONCLUSIONS

In conclusion, we introduce UnrealZoo which offers a versatile platform for advancing embodied AI research. The diverse, realistic complex environments challenge agents with varying tasks such as visual navigation, active tracking across various environments, and social tracking in crowds. The enhanced UnrealCV+ API supports efficient data collection, customization, and task creation, enabling seamless interaction for both single and multi-agent systems. These features will open up potential applications like multi-agent rescue missions, collaborative searching, and industrial automation, making our platform a valuable tool for pushing the boundaries of embodied AI in real-world scenarios.

**Limitations.** While our proposed environment provides diverse and complex scenarios for visual navigation, tracking, and other visual-based tasks, it currently lacks high-fidelity physical simulation, limiting the agent's ability to interact with objects. The interaction between the agent and tiny objects, such as manipulating objects, is also minimal. Additionally, transferring learned behaviors to different embodied agents poses a challenge, as adapting models to various physical structures and control schemes is not yet seamless. These issues highlight areas for further research to enhance interaction dynamics and improve generalization across diverse agent embodiments.

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

# A  UE ENVIRONMENTS

## A.1  COMPARISON WITH OTHER SIMULATORS

To better explain Table 1, we list the description of each symbol about the scene types and playable entities in Table 6. Since photorealism mainly relies on the engine used, we visualize the snapshots rendered by different engines in Figure 6. Note that Google Maps are images captured in the real world, but can not simulate the dynamic of the scenes and interactions between objects. By utilizing advanced rendering and physics engines, Unreal Engine simulates large-scale photorealistic environments that are not only visually appealing but also capable of complex interactions between agents and objects. So we choose to build environments on Unreal Engine.

Table 6: The description of symbols used in Table 1.

| Symbol | Description |
|---|---|
| | Scenes with indoor furnishings |
| | Scenes with outdoor roads |
| | Natural landscapes with trees of varying heights and grasslands |
| | Buildings with castle-style architecture |
| | Realistic construction site scenes with a variety of construction tools and equipment |
| | Realistic factory scenes with internal roads and factory facilities |
| | Scenes with residential community settings |
| | Scenes featuring temple architecture with stairs, lofts, and shrines |
| | Sports venue scenes |
| | Common urban public transportation station scenes |
| | Hospital interior scenes with detailed elements |
| | High-fidelity urban environments |
| | Scenes with a desert and seaside landscape style |
| | Human agents with detailed features such as hair textures, clothing, and actions |
| | Mobile robot |
| | Driveable car |
| | Animal agents, including common animal species such as cats, dogs, lions, tigers, etc. |
| | Driveable motorbike |
| | Drones |
| | Virtual camera that has no physical entity and is movable |

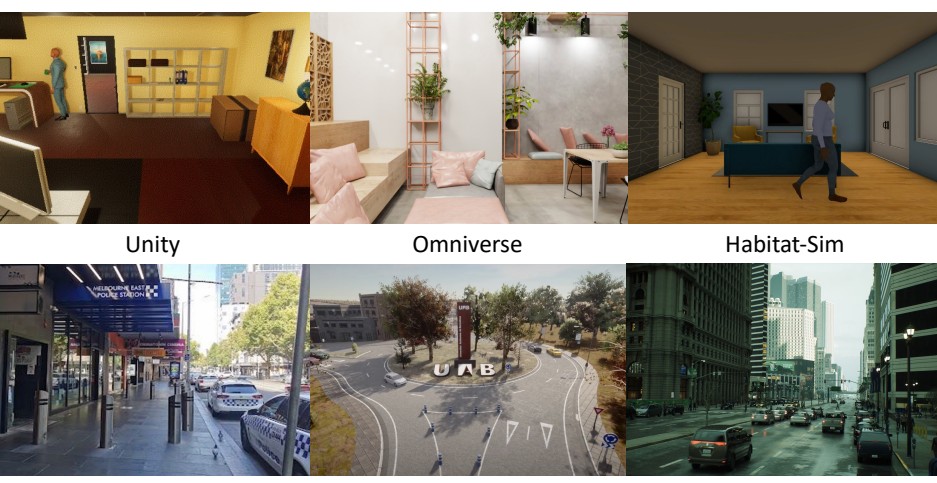

|  |  |  |
|---|---|---|
| Unity | Omniverse | Habitat-Sim |
| Google Map | Unreal Engine 4 | Unreal Engine 5 |

Figure 6: Comparison of the visual realism of different engines: we show the snapshots captured from different engines to compare the photo-realism of different environments for an intuitive feeling. Note that Google Maps capture and reconstruct the images from the real world, but can not simulate the dynamic of the scenes and interactions between agents and objects.

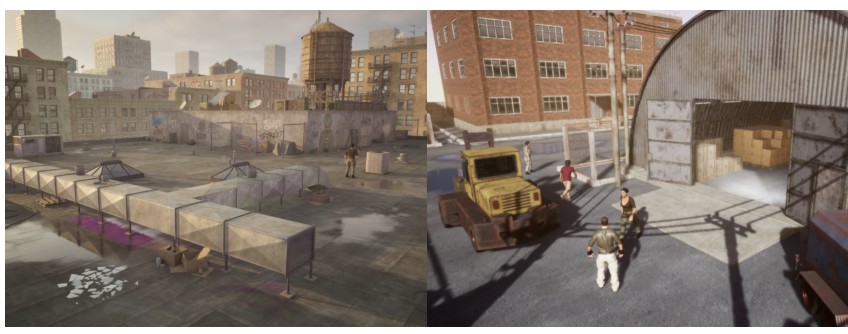

Roof             Factory

Figure 7: Two photo-realistic environments used for visual navigation.

## A.2 ENVIRONMENTS USED IN VISUAL NAVIGATION

We carefully selected two photo-realistic environments (**Roof** and **Factory**) for training and evaluating navigation in the wild, shown in Figure 7. The Roof environment features multiple levels connected by staircases and large pipelines scattered on the ground, providing an ideal setting for the agent to learn complex action combinations for transitioning between levels, such as jumping, climbing, and navigating around obstacles. The Factory environment, on the other hand, is characterized by compact boxes and narrow pathways, challenging the agent to determine the appropriate moments to jump over obstacles or crouch to navigate under them. These two environments offer diverse spatial structures, enabling agents to develop an understanding of multi-level transitions and precise obstacle avoidance.

## A.3 ENVIRONMENTS USED IN ACTIVE VISUAL TRACKING

For training agents via offline reinforcement learning, we selected 8 distinct environments to collect demonstrations, as is shown in Figure 10. To comprehensively evaluate the generalization of the active visual tracking agents, we selected **16** distinct environments, categorized into Interior Scenes, Palaces, Wilds, and Modern Scenes. Each category presents unique challenges: 1) **Interior Scenes** feature complex indoor structures with frequent obstacles; 2) **Palaces** include multi-level structures and narrow pathways; 3) **Wilds** encompass irregular terrain and varying illumination; 4) **Modern Scenes** offer high-fidelity, real-world scenarios with modern buildings and objects. These diverse environments facilitate a thorough assessment of the agent's generalization capabilities across varying complexities. The snapshot of each environment is shown in Figure 8.

## A.4 NAVIGATION MESH

Based on NavMesh, we build an internal navigation system, allowing agents to autonomously navigate with the built-in AI controller in the Unreal Engine. This includes path-finding and obstacle-avoidance capabilities, ensuring smooth and realistic movement throughout diverse terrains and structures. Moreover, in our City style map, we manually construct road segmentation, we manually segment the roads to distinguish between pedestrian and vehicle pathways. When agents use the navigation system for autonomous control, they will navigate the shortest path based on the priority of the different areas. Figure 9 shows an example of the rendered semantic segmentation for NavMesh in an urban city.

## B EXEMPLAR TASKS

### B.1 VISUAL NAVIGATION

In this task, the agent is initialized at a random location in the environment at the beginning of each episode, while the target object's location and category remain fixed throughout. The agent must rely on its first-person view observations and the relative spatial position of the target as input. The

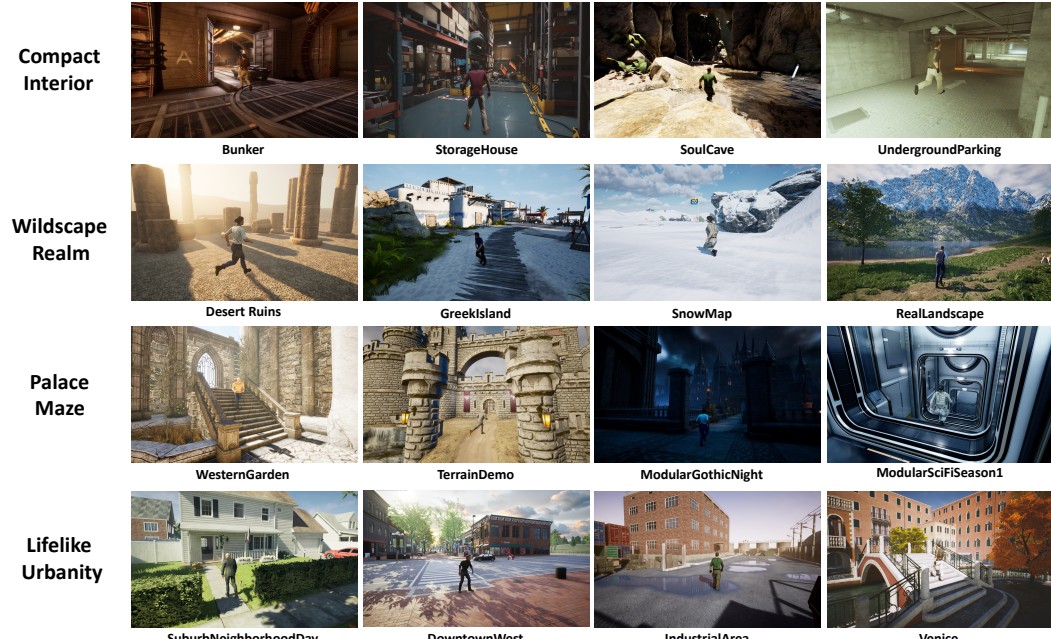

Figure 8: The snapshots of 16 environments used for testing active visual tracking agents. The text on the left indicates the category corresponding to that line of environment.

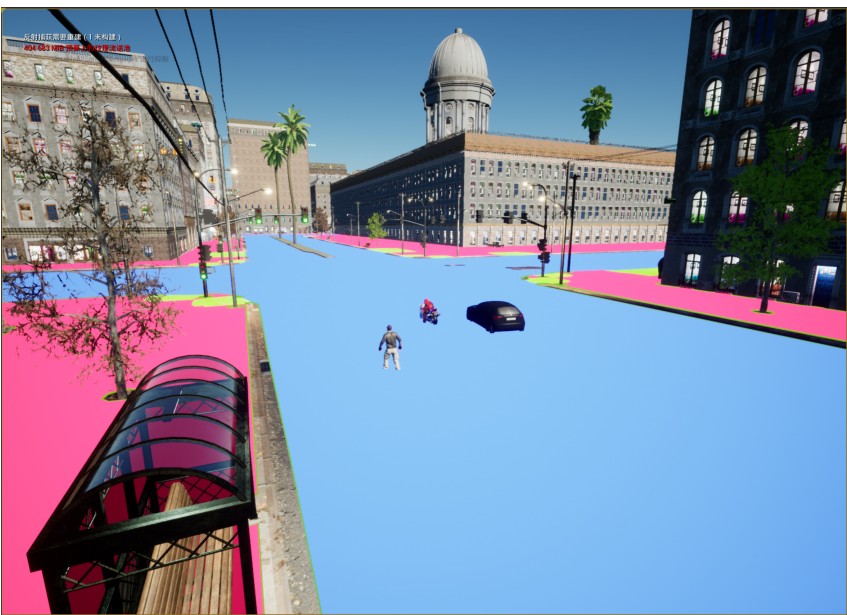

Figure 9: An example of the NavMesh with semantic segmentation. The human character will prioritize using the pink area for pedestrian navigation tasks, while the vehicles will use the blue area.

ultimate objective is to locate the target object within 2000 steps. Success is defined by the agent reducing the relative distance to less than 3 meters and aligning its orientation such that the relative rotation between the target and the agent is smaller than 30 degrees (in the front of the agent). This setup challenges the agent to optimize its movements and decision-making while adapting to the randomized starting conditions and dynamic environment. All methods in the task share the same discrete action space to control the movement, consisting of moving forward (+1 meter/s), moving backward (-1 meter/s), turning left (-15 degrees/s), turning right (+15 degrees/s), jumping (two

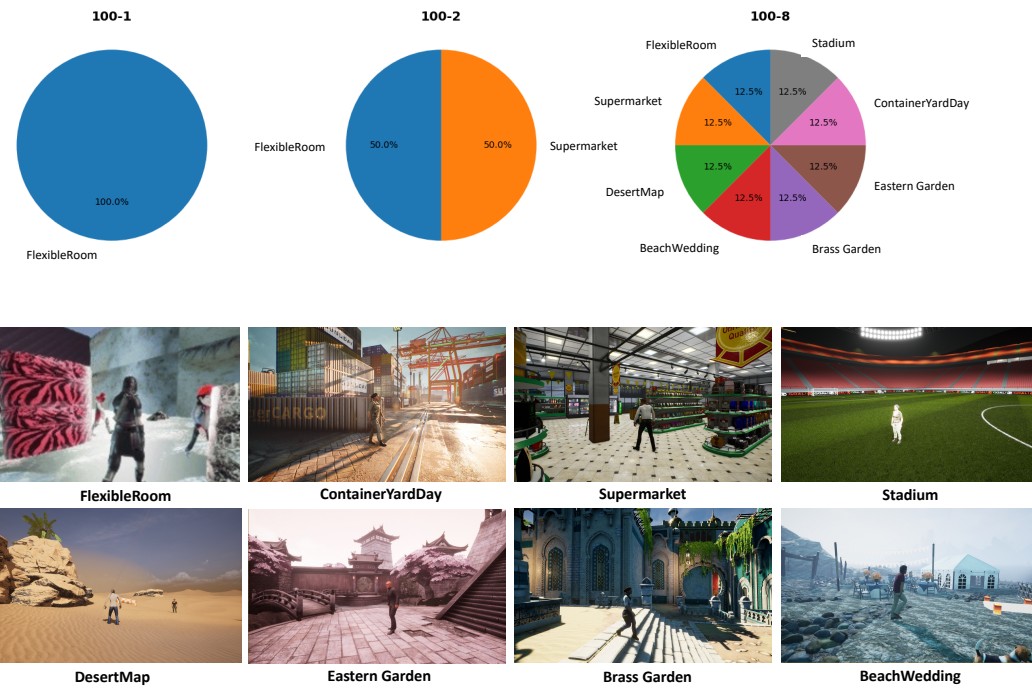

Figure 10: The 8 environments used for collecting offline dataset.

continuous jumping actions trigger the climbing action), crouching, and holding position. This action space enables the agent to navigate and interact with complex 3D environments, making strategic decisions in real-time to reach the target object efficiently. The step reward for the agent is defined as:

$$r(t) = \tanh(\frac{dis2target(t-1) - dis2target(t)}{max(dis2target(t-1), 300)} - \frac{|Ori|}{90°}) \tag{1}$$

where $dis2target(t)$ is the Euclidean distance between the agent and the target at a given timestep $t$ and $|Ori|$ is the absolute orientation error (in degrees) between the agent's current heading and the direction toward the target, normalized by $90°$

### B.2 ACTIVE VISUAL TRACKING

Referring to previous works (Zhong et al., 2024), we use human characters as an agent player and a continuous action space for agents. The action space contains two variables: the angular velocity and the linear velocity. Angular velocity varies between $-30°/s$ and $30°/s$, while linear velocity ranges from $-1\,m/s$ to $1\,m/s$. In the agent-centric coordinate system, the reward function is defined as:

$$r = 1 - \frac{|\rho - \rho^*|}{\rho_{max}} - \frac{|\theta - \theta^*|}{\theta_{max}} \tag{2}$$

where $(\rho, \theta)$ denotes the current target position relative to the tracker, $(\rho^*, \theta^*) = (2.5m, 0)$ represents the expected target position, i.e., the target should be $2.5m$ in front of the tracker. The error is normalized by the field of the view $(\rho_{max}, \theta_{max})$. During execution, an episode ends with a maximum length of 500 steps, applying the appropriate termination conditions. In the experiment, we adopt the original neural network structure and parameters, as listed in Table 9 and 10.

### B.3 TASK CONFIGURATION IN A JSON FILE

We provide an example of the task configuration JSON file in Figure 11. Using the JSON file, we can easily set the configuration of the binary, the continuous and discrete action space for each agent, the placement of the binding camera, choose the area to reset, and other hyper-parameters about the environments.

```
A Json File for Task Configuration

    "env_name": env_name,
    "env_bin":path-to-binary,
    "env_map": map_name,
    "env_bin_win": path-to-binary(for windows),
    "third_cam": {"cam_id": 0,"pitch": -90,"yaw": 0,"roll": 0,"height_top_view":
        1460.0,"fov": 90},
    "height": 460.0,
    "interval": 1000,
    "agents": {
        "player": {
            "name": ["BP_Character_923"],
            "cam_id": [3],
            "class_name": ["bp_character_C"],
            "internal_nav": true,
            "scale": [1,1,1],
            "relative_location": [20,0,0],
            "relative_rotation": [ 0,0,0],
            "head_action_continuous": {"high": [15,15,15], "low": [-15,-15,-15]},
            "head_action": [ [0,0,0],[0,30,0],[0,-30,0]],
            "animation_action": ["stand","jump","crouch"],
            "move_action": [
            [angular, velocity]
                ...
            ],
            "move_action_continuous": {"high": [30,100],"low": [-30,-100]}
        },
        "animal": {
            "name": ["BP_animal_2"],
            "cam_id": [1],
            "class_name": ["BP_animal_C"],
            "internal_nav": true,
            "scale": [1,1,1],
            "relative_location": [20,0,0],
            "relative_rotation": [0,0,0],
            "move_action": [
                [angular, velocity]
                ...
            ],
            "move_action_continuous": { "high": [30,100],"low": [-30,-100]}
        },
        "drone": {
            "name": ["BP_Drone01_2"],
            "cam_id": [2],
            "class_name": ["BP_drone01_C"],
            "internal_nav": false,
            "scale": [ 0.1,0.1,0.1],
            "relative_location": [0,0,0],
            "relative_rotation": [0,0,0],
            "move_action": [
              [angular, velocity]
                ...
            ],
            "move_action_continuous": {"high": [1,1,1,1],"low": [-1,-1,-1,-1]}
        }
    },
    "safe_start": [
        [x,y,z],
        ...
    ],
    "reset_area": [x_min,x_maxin,y_min,y_max,z_min,z_max],
    "random_init": false,
    "env": {"interactive_door": []},
    "obj_num": 466,
    "size": 192555.0,
    "area": 9900.0,
    "bbox": [110.0, 90.0,19.45]
```

Figure 11: An example of the task configuration file in JSON format.

### B.4 COLLECTING DEMONSTRATION FOR ACTIVE VISUAL TRACKING

To demonstrate the flexibility of the environment, we use state-based expert policy and the multi-level perturbation strategy (Zhong et al., 2024) to automatically generate various imperfect demonstrations

as the offline dataset. For active visual tracking, we employ three distinct datasets for training agents via offline reinforcement learning (Offline RL) algorithms, referred to as *1 Env.*, *2 Envs.*, and *8 Envs*. The detailed composition of each dataset is depicted in Figure 10. For the *1 Env.* dataset, we use only the FlexibleRoom, an abstract environment enriched with diverse augmentation factors, to gather 100k steps of trajectory data. For 2 Envs., we collect 50k step trajectories from FlexibleRoom and an additional 50k steps from the Supermarket environment. The 8 Envs. dataset involves eight different environments, with 12.5k steps collected from each. Therefore, **the total amount of data in the three datasets is the same (100k) to ensure the fairness of the comparison.** These dataset configurations aim to highlight the critical role of environment diversity in enhancing the generalization capabilities of embodied AI agents.

## C  IMPLEMENTATION DETAILS OF AGENTS

### C.1  RL-BASED AGENTS

**Learning to navigate with online reinforcement learning.** For navigation, we construct an RL-based end-to-end model, using A3C (Mnih et al., 2016) to accelerate online reinforcement learning in a distributed manner. The model's structure is as follows: a mask encoder extracts spatial visual features from the segmentation mask, which are then passed to a temporal encoder to capture latent temporal information. Finally, the spatiotemporal features, concatenated with the target's relative spatial position, are fed into the actor-critic network to optimize the actor layer for action prediction. The detailed network structure and parameters used in the experiment are listed in Table 7 and 8. Here, we provide the training curves in *Roof* and *Factory* environments, depicted in Figure 15. In the *Factory*, we set the number of workers to 4, while in the *Roof*, the number of workers is set to 6. It can be observed that, for Online RL, the number of workers and the complexity of environments have a significant impact on training efficiency. Looking forward, we anticipate that offline-based algorithms can effectively address the challenges of training efficiency and generalization.

Table 7: Details the neural network structure of RL-based agent for navigation task, where 5×5-32S1 means 32 filters of size 5×5 and stride 1, FC256 indicates the fully connected layer with output dimension 256, and LSTM128 indicates that all the sizes in the LSTM unit are 128.

| Module | Mask Encoder | | | | | | | |
|---|---|---|---|---|---|---|---|---|
| Layer# | CNN | Pool | CNN | Pool | CNN | Pool | CNN | Pool |
| Parameters | 5×5-32S1 | 2-S2 | 5×5-32S1 | 2-S2 | 4×4-64S1 | 2-S2 | 3×3-64S1 | 2-S2 |

| Module | Temporal Encoder | | Actor | Critic |
|---|---|---|---|---|
| Layer# | FC | LSTM | FC | FC |
| Parameters | 256 | 128 | 2 | 2 |

Table 8: The experiment setting and hyper-parameters used for training the RL-based navigation agent.

| Name | Value | Name | Value |
|---|---|---|---|
| Learning Rate | 1e-4 | LSTM update step | 20 |
| workers (Roof) | 6 | LSTM Input Dimension | 256 |
| workers (Factory) | 4 | LSTM Output Dimension | 128 |
| Position Input Dimension | 2 | LSTM Hidden Layer size | 1 |

**Learning to track with offline reinforcement learning.** For the tracking task, we adopt an offline reinforcement learning (Offline RL) approach to enhance training efficiency and improve the agent's generalization to unknown environments. Specifically, we build an end-to-end model trained using offline data and the conservative Q-learning (CQL) strategy (Kumar et al., 2020). We adopt the same model structure from the latest visual tracking agent (Zhong et al., 2024), consisting of a Mask Encoder, a Temporal Encoder, and an Actor-Critic network. Detailed model structures and training parameters are summarized in Table 9 and 10. Additionally, we provide the model's loss curves under different dataset setups, as shown in Figure 14. The model achieves near-convergence within

two hours across all dataset setups. To ensure the loss curves stabilize fully, we continued training for an additional three hours, during which no significant further decrease in the loss was observed. A comprehensive evaluation of the model's performance is presented in Tables 11 and 12, highlighting its strong generalization to unseen environments and robustness to dynamic disturbances. The training efficiency, generalization capability, and robustness achieved by offline RL further reinforce our belief that offline RL methods will become a mainstream approach for rapid prototyping and iteration in embodied intelligence systems.

Table 9: Network structure used in the offline RL method (Zhong et al., 2024), where 8×8-16S4 means 16 filters of size 8×8 and stride 4, FC256 indicates a fully connected layer with dimension 256, and LSTM64 indicates that all sizes in the LSTM unit are 64.

| Module | Mask Encoder | | | Temporal Encoder | Actor | Critic |
|---|---|---|---|---|---|---|
| Layer# | CNN | CNN | FC | LSTM | FC | FC |
| Parameters | 8×8-16$S4$ | 4×4-32$S2$ | 256 | 64 | 2 | 2 |

Table 10: The hyper-parameters used for offline training and the policy network.

| Name | Value | Name | Value |
|---|---|---|---|
| Learning Rate | 3e-5 | LSTM update step | 20 |
| Discount Factor | 0.99 | LSTM Input Dimension | 256 |
| Batch Size | 32 | LSTM Output Dimension | 64 |
| LSTM Hidden Layer size | 1 | | |

## C.2 VLM-BASED AGENTS

We built agents with a reasoning framework based on the Large Vision-Language Model. We employ OpenAI GPT-4o as the base model. System prompt used in the navigation task, as shown in Figure 13 and system prompt used in the tracking task, as shown in Figure 12.

## C.3 HUMAN BENCHMARK FOR NAVIGATION

In the navigation task, we incorporated human evaluation as a baseline for comparison to demonstrate the existing gap between the current method and optimal navigation performance. Specifically, **five male and five female** evaluators participated in the assessment, performing the same navigation tasks under comparable conditions.

Before each human evaluator began their assessment, we provided a free-roaming perspective to familiarize them with the map structure and clearly conveyed the target's location and image. This ensured that human evaluators had a comprehensive understanding of the environment and the target's position. During the evaluation, the player was randomly initialized in the environment, and human evaluators used the keyboard to control the agent's movements. Each human evaluator repeated the experiment five times, providing multiple data points to ensure reliability and reduce variability in performance measurements. The termination conditions for the evaluation were identical to those applied to the RL-based agent, ensuring consistency in the comparison.

**System Prompt used for active tracking**

```
Objective:
You are an intelligent tracking agent designed to control the robot to track the person
    in the view. The first person in your view is your target. You need to provide
    concrete moving strategie to helo robot tracking the target in the given
    environment.

Representation details:
1. Moving instructions are concrete actions that the robot can take to adjust its
    viewpoint and distance to the target. The moving instructions include:
    -move closer: Move the robot closer to the target. This should be chosen when the
        target is too far away from the robot and there is no obstacle in the way.
    -move further: Move the robot further away from the target. This should be 2chosen
        when the target is too close to the robot and only part of the target body is
        visible in the view.
    -keep current: Maintain the current distance and angle between the robot and the
        target. This is chosen when the target is fully observable in the view and
        there is enough space in front of both tracker and target without any
        potential obstacles may cause collision and occlusion.
    -turn left:  Turn the robot to left direction, the target will move towards the
        right side in next frame.
    -turn right: Turn the robot to right direction, the target will move towards the
        left side in next frame.
Input Understanding:
1.**Image:** We provide a first-person view observation of the robot to help you
    understand the surrounding environment. The observation is represented as a color
    image from the tracker's first-person perspective.

Output Understanding:
1. **Moving Strategy:** A temporal reasonable move strategy to adjust the robot
    viewpoint and distance to achieve robots's long-term tracking task. This should be
     represented as a concrete moving instructions, the instructions should be choose
    from "move closer", "move further" ,"keep current", "turn left","turn right".
    Format - [Keep current].

Strategy Considerations:
1.If the person's horizontal position in the robot's field of view deviates from the
    center by more than 25% of the image width, we consider the target to be on one
    side of the image, otherwise we say the target is near the center.
2.To provide a reasonable moving strategy, you should think step by step based on the
    input image and the following hints:
    1)If the person is too close to the robot and the target in the image is clipped,
        robot should move further first to obtain a better view.
    2)If the person's size in the view is too small in the image, robot should move
        closer to obtain a better view.
    3)If the person may occluded by obstacles or structures in the future, the robot
        should move closer to avoid losing the person in the next frame.
    4)If the person is near the right edge in the image and there is no immediate
        obstacle in front of robot, the robot should turn right to keep person near
        center in the image.
    5)If there is immediate hinder obstacles in front of the robot, turn right or left
        to a clean space first.
    6)If there is any potential occlusion effect or obstacles on either side of the
        person's walking path, the robot should move closer to avoid losing the person
         in the next frame.
    7)If there is no person in the current image, turn right or turn left to search the
         person.
Instructions:
1.Provide ONLY the decision in the [output:] strictly following the format without
    additional explanations or additional text.
```

Figure 12: System prompt used for tracking.

**System Prompt used for navigation**

```
Objective:
You are an intelligent navigation agent designed to control the robot to navigate to
    the target object location based on first-person observation and provide a
    relative position between the robot and the target. You need to provide an action
    sequence to help the robot move to the target location.

Representation details:
1. Relative Position: This contains three elements, in the format - [Distance,
    Direction, Height].
    -Distance: The relative distance between the robot and the target object.
    -Direction: The target object's relative direction to the robot, represented in
        degrees. \
            A positive value represent the target is on the right side of the robot
                with corresponding angle and a negative value represent the target is
                on the left side of the robot with corresponding angle. \
            The absolute value of the angle larger than 90 degree means the target is
                behind the robot. \
    -Height: The relative vertical position, where a positive value indicates that the
        target is higher than the robot.
1. Actions: These are the movements the robot can perform to adjust its position. The
    available actions include:
    -Move Forward: Propel the robot forward by 100 centimeter.
    -Move Backward: Propel the robot backward by 100 centimeter.
    -Turn Left: Rotate the robot 15 degrees to the left.
    -Turn Right: Rotate the robot 15 degrees to the right.
    -Jump: Make the robot leap into the air, robot should use this action to jump over
        obstacles or climb over stairs.
    -Crouch: Lower the robot into a crouching position for 2 seconds, after which it
        will automatically stand up.
    -Keep Current: Maintain the robot's current position without any movement.

Input Understanding:
1.**Image:** We provide a first-person view observation of the robot to help you
    understand the surrounding environment. The observation is represented as a color
    image from the robot's first-person perspective.
2.**Relative Position:** This data provides the target object's relative position to
    the robot, including the distance, direction, and height. The distance is measured
    in centimeters, the direction in degrees, and the height in centimeters.
Output Understanding:
1. **Action Sequence:** This is a series of Three continuous actions that the robot
    should take to navigate toward the target object. Each sequence must consider the
    provided relative position data and the first-person observation. \
                        The actions should be ordered logically to effectively move the
                            robot closer to the target, adjusting its direction,
                            distance, and height as needed. \
                        The action sequence should be clear and executable, enabling
                            the robot to reach the target efficiently while avoiding
                            obstacles and maintaining stability
                        in the format - [Action1, Action2, Action3]. Each action should
                            be choose from the available actions mentioned above.

Strategy Considerations:
1.Assessing Relative Position: Begin by evaluating the target object's relative
    position in terms of distance, direction, and height to inform the action sequence
    .
2.Action Combination for Navigation: Utilize the action sequence to create effective
    combinations, each action will last for 1 seconds. For example:
        -Consider using multiple consecutive actions like [Move Forward, Jump, Jump] to
            climb over the front obstacles or boxes.
        -Consider using [Move Backward,Move Backward,Move Backward] to move the robot
            avoid a front wall or fence.
3.Obstacle Detection: Leverage the first-person observation to identify obstacles.
    Based on their location, formulate action sequences that facilitate smooth
    navigation while avoiding collisions.
4.Efficient Pathing: Ensure the action sequence is designed to dynamically adjust the
    robot movement torward target object, which is minimize the distance and direction
     value in **Relative Position**.
5.Sequence Validation: Validate the generated action sequence and consider past
    memories to ensure it is practical given the current environment and obstacles,
    making long-term adjustments as necessary.
Instructions:
1.Provide ONLY the action sequence in the [output:] strictly following the format -[
    Action1, Action2, Action3], without additional explanations or additional text.
```

Figure 13: System prompt used for navigation.

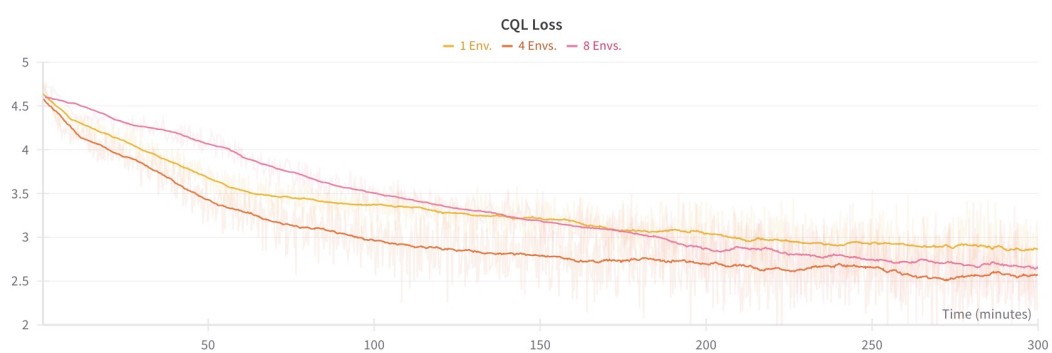

Figure 14: The CQL loss curve during offline training with different offline datasets.

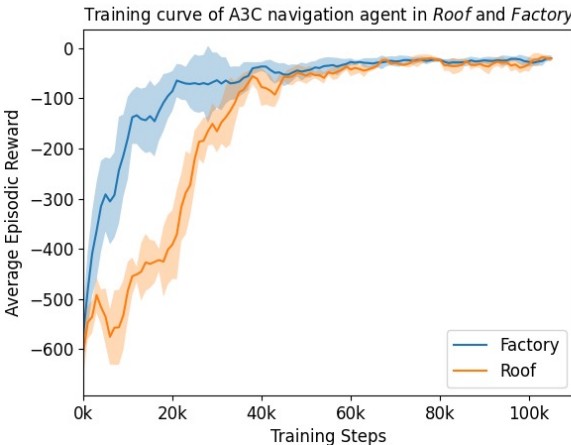

Figure 15: The learning curves for RL-based navigation agent in two environments: Roof and Factory. We use A3C (Mnih et al., 2016) to learn the navigation policy via trial-and-error interactions. In the Factory (blue line plot), the number of asynchronous workers is set to 4, while in the Roof environment (orange line plot), the number of asynchronous workers is set to 6.

# D    ADDITIONAL RESULTS

## D.1    LEARNING CURVE

We provide the CQL loss curve under the *1 Env., 4 Envs. and 8 Envs.* training setup. As shown in Figure 14, the offline model approaches convergence after two hours and we continued training for another three hours after nearing convergence, observing no significant further decrease in the loss. Note that the offline training was conducted on a Nvidia RTX 4090 GPU.

## D.2    EVALUATE TRACKING AGENTS ACROSS 16 UNSEEN ENVIRONMENTS

We provide the detailed quantitative evaluation results (episodic returns, episode length, success rate) of the RL-based embodied tracking agents across 16 environments, listed in Table 11. In each environment, we report the average results over 50 episodes. The results show that in the *Palace Maze*, which contains abundant structural obstacles, the agent's tracking performance was generally weaker compared to the other three categories. In contrast, the agent performed generally better in *Lifelike Urbanity*, characterized by its relatively regular and flat terrain. Additionally, we observed that as the diversity of the training environments increased, the agent's tracking performance improved across all four environment categories. This highlights the positive impact of diverse training data on enhancing the agent's overall tracking effectiveness. We also provide vivid demo videos in `https://unrealzoo.notion.site/task-evt`.

Table 11: Quantitative evaluation results of the offline RL method across 16 environments. The environments are grouped into four categories: Compact Interior, Wildscape Realm, Palace Maze, and Lifelike Urbanity. The table compares the performance of agents trained on different offline dataset settings: 1 Env. (single environment), 2 Envs. (two environments), and 8 Envs. (eight environments). Each cell presents three metrics from left to right: Average Episodic Return (ER), Average Episode Length (EL), and Success Rate (SR).

| Category | Environment Name | 1 Env. ER/EL/SR | 2 Envs. ER/EL/SR | 8 Envs. ER/EL/SR |
|---|---|---|---|---|
| Compact Interior | Bunker | 241/412/0.56 | 245/391/0.56 | 234/429/0.70 |
| | StorageHouse | 213 /424 /0.68 | 275/449/0.76 | 170/434/0.64 |
| | SoulCave | 229/402/0.60 | 252/422/0.56 | 206/405/0.58 |
| | UndergroundParking | 179/391/0.56 | 250/424/0.62 | 184/410/0.60 |
| Wildscape Realm | Desert Ruins | 209/392/0.54 | 293/449/0.70 | 277/453/0.70 |
| | GreekIsland | 245/411/0.62 | 264/423/0.64 | 257/466/0.78 |
| | SnowMap | 204/399/0.62 | 322/456/0.78 | 278/474/0.86 |
| | RealLandscape | 171 /383/0.42 | 225/372/0.44 | 223/444/0.70 |
| Palace Maze | WesternGarden | 230/403/0.54 | 209/408/0.54 | 296/472/0.82 |
| | TerrainDemo | 232/411/0.56 | 233/403/0.56 | 192/411/0.56 |
| | ModularGothicNight | 190/360/0.52 | 244/423/0.62 | 272/456/0.76 |
| | ModularSciFiSeason1 | 168/365/0.42 | 172/354/0.42 | 211/393/0.48 |
| Lifelike Urbanity | SuburbNeighborhoodDay | 224/422/0.64 | 328/457/0.72 | 242/457/0.76 |
| | DowntownWest | 296/460/0.78 | 317/456/0.76 | 292/469/0.86 |
| | Factory | 278/434/0.64 | 291/452/0.74 | 249/435/0.64 |
| | Venice | 295/441/0.70 | 323/448/0.82 | 294/474/0.84 |

Table 12: Quantitative evaluation results of the tracking agents across 4 different category environments with **4 distractors (4D), 8 distractors (8D), and 10 distractors (10D)** respectively. The table compares the performance of agents trained on different offline dataset settings: 1 Env. (single environment), 2 Envs. (two environments), and 8 Envs. (eight environments). Each cell presents three metrics from left to right: Average Episodic Return (ER), Average Episode Length (EL), and Success Rate (SR).

| Category | Environment Name | 1 Env. ER/EL/SR | 2 Envs. ER/EL/SR | 8 Envs. ER/EL/SR |
|---|---|---|---|---|
| Compact Interior | StorageHouse (4D) | 117/343/0.40 | 181/375/0.52 | 190/428/0.62 |
| | StorageHouse (8D) | 143/341/0.34 | 151/338/0.44 | 165/366/0.49 |
| | StorageHouse (10D) | 81/324/0.36 | 109/331/0.42 | 107/357/0.50 |
| Wildscape Realm | DesertRuins (4D) | 317/469/0.72 | 333/456/0.70 | 354/466/0.74 |
| | DesertRuins (8D) | 213/406/0.50 | 316/445/0.58 | 267/444/0.68 |
| | DesertRuins (10D) | 188/390/0.44 | 252/382/0.50 | 253/447/0.64 |
| Palace Maze | TerrainDemo (4D) | 221/398/0.44 | 286/454/0.65 | 312/460/0.77 |
| | TerrainDemo (8D) | 211/384/0.39 | 239/412/0.49 | 252/420/0.52 |
| | TerrainDemo (10D) | 189/377/0.36 | 232/404/0.48 | 224/429/0.66 |
| Lifelike Urbanity | SuburbNeighborhoodDay (4D) | 192/407/0.46 | 256/381/0.50 | 265/392/0.60 |
| | SuburbNeighborhoodDay (8D) | 131/325/0.36 | 229/369/0.48 | 247/385/0.56 |
| | SuburbNeighborhoodDay (10D) | 162/355/0.44 | 180/340/0.40 | 165/376/0.44 |

## D.3 EVALUATE TRACKING AGENTS ACROSS UNSEEN SOCIAL ENVIRONMENTS

We select 4 environments from different categories as the testing environments, including Storage-House, DesertRuins, TerrainDemo, and SurburNeighborhoodDay. We test the distraction robustness of the social tracking agents by adding different numbers of distractors (4, 8, 10) in the environment. The distractors randomly walk around the environment, which may produce various unexpected

perturbations to the tracker, such as visual distractions, occlusion, or blocking the tracker's path. As shown in Table 12, the tracking performance of the three agents steadily decays with the increasing number of distractors.

