# OpenReview forum: "UnrealCV Zoo: Enriching Photo-realistic Virtual Worlds for Embodied AI Agents"
_ICLR.cc/2025/Conference — Submitted to ICLR 2025_

### Official Review · Reviewer_u5UD · 2024-10-22

**Soundness:** 2
**Presentation:** 3
**Contribution:** 2
**Rating:** 6
**Confidence:** 2

**Summary:**

The paper proposes a photo-realistic simulation environment for embodied training and a scene dataset of 100 diverse scenes.
The simulator is based on UnrealCV develops interfaces for RL training, and supports multi-agent.
The simulator highlights its simulation speed and diversity of scenes and entities. Necessary experiments are conducted to show its usability.

**Strengths:**

- The environment is photo-relaisitic.
- Scene quality an is good. Entities are diverse. Simulation efficiency is improved.
- There are a lot of experiments.

**Weaknesses:**

- There are only 100 scenes and they are not scalable.
- There is no comparison with more recent related works like [1][2].
- The motivation is not clear in introduction. What's the disadvantages of previous simulators? How does UNREALZOO settle them?
- The interaction between agents and the world is not discussed, which is important for embodied training.
- There are no language-modal input/communication among agents. The multi-agent feature is not highlighted.


[1] Cheng, Zhili et al. “LEGENT: Open Platform for Embodied Agents.” ArXiv abs/2404.18243 (2024): n. pag.

[2] Yang, J., Ding, R., Brown, E., Qi, X., & Xie, S. (2024). V-irl: Grounding virtual intelligence in real life. arXiv preprint arXiv:2402.03310.

**Questions:**

- About the choice of base engine, is UE the only engine that can render photo-realisitic images?
- There is a lot of Agent body mentioned in table 1. Are they driven by pre-defined policy? What are their action spaces?
- Cannot find Table 3.3 in line 308 and Table 4.1 in line 368. `GPT4-o `in table 5 should be `GPT-4o`.
- Why GPT-4o performs so poorly? Do you do the error breakdown analysis? Can you use other information like position or third-person view to increase the success rate?
- In social tracking experiment, as the training set grows, the growth of SR is intuitive. Can you conduct the experiment on same scale training (compared to the less diverse data) to give a fair evaluation?

**Details Of Ethics Concerns:**

There may be copyright problem of using scenes in the market.

---

> ### Author Response · Authors · 2024-11-26
> **Response to Reviewer u5UD (1)**
>
> Thank you for your valuable feedback. Below, we address each of your points in detail.
>
> > **W1:** There are only 100 scenes and they are not scalable.
>
> **R1**: We do not agree with that at all. It is unfair to simply compare the number of scenes at different scale level. The scale level and diversity of our collected environments are significantly greater than in previous indoor environments.
> For example, there are numerous houses and rooms in a town-level scene, such as [SuburbNeighborhoodDay](https://unrealzoo.notion.site/SuburbNeighborhood_Day-4eab6005bf1e46418e1860eb6c85897f). The largest scene ([Medieval Nature Environment](https://unrealzoo.notion.site/Medieval-Nature-Environment-0706190c7b2641a98de714bf546f7e0a?pvs=25)) covers more than $16 km^2$ areas. As is shown in Table 1, **the scenes collected in UneralZoo are of much greater diversity in scene types, scale, and playable entities than previous works.**
>
> Regarding **scalability**, we provide a set of functions to manipulate the scene, including `spawning new objects`, `destroying objects`, `setting object locations and rotations`, and `rescaling object sizes`. Users can easily customize new scenes based on our pre-built environments. The illumination, camera parameters, the texture of objects, and animations of players can also be controlled by UnrealCV+. Moreover, we will also provide detailed tutorials to help users package new environments with UnrealCV+, fostering community contributions. Hence, most environments in [Fab](https://www.fab.com/) can be added to UnrealZoo with minimal effort.
>
> >**W2:** There is no comparison with more recent related works like [1][2].
>
> **R2**: Thank you for your suggestion. In the revision, we have included two related works in `Table 1` for comparison. Specifically, LEGENT[1] focuses on multi-modal interactions between agents and objects, offering only two pre-built scenes. There is only a virtual human character and a robot-like character in the scene. It is built on Unity, and the communication code [has not been open-sourced](https://github.com/thunlp/LEGENT/issues/9) yet. In contrast, Unreal Engine provides more advanced rendering, and UnrealCV has been open source for years. Our extension (UnrealCV+) will also be open-sourced upon the official public release of the environments. V-IRL[2] utilizes Google Maps APIs to allow agents to navigate in reconstructed large-scale open worlds. However, it only includes static outdoor scenes, and the agents cannot interact physically with any objects.
>
> >**W3:** The motivation is not clear in introduction. What's the disadvantages of previous simulators? How does UnrealZoo settle them?
>
> **R3**: Our ultimate goal is to simulate complex open worlds to build generalist embodied agents for real-world applications. The main limitation of previous simulators is their **lack of diversity in scenes and characters**. Many focus on specific scenarios, such as daily home activities or urban autonomous driving, which hinders the advancement of generalist embodied AI in open worlds. For example, simulators like AI2-THOR, OmniGibson, VirtualHome, and Habitat primarily emphasize indoor interactions for household robots. In contrast, UnrealZoo provides a collection of high-quality, photo-realistic, large-scale scenes that encompass a variety of environments, thereby offering a more comprehensive research platform. Additionally, UnrealZoo features diverse playable entities, including humans, animals, robots, and drones, facilitating research into cross-embodiment generalization and heterogeneous multi-agent interactions.
>
> Minecraft is a choice to simulate the open worlds but the photorealism is limited. The photorealism of the environment is required to reproduce the real-world challenges to train and evaluate the embodied visual agents for real-world applications. With advancements in Unreal Engine, UnrealZoo achieves photorealistic rendering, enhancing the realism of the simulated environments to mitigate the sim2real gaps.
>
> >**W4:** The interaction between agents and the world is not discussed, which is important for embodied training.
>
> **R4**: We appreciate your concern regarding the diversity of interactions and actions in our environments. However, we would like to clarify that our action space already supports a broad range of interactions, such as picking up objects, climbing obstacles, opening and closing doors, entering and exiting vehicles (cars or motorbikes), falling, crouching, PTZ control, and so on. We provide video demonstrations of some typical actions on [this website](https://unrealzoo.notion.site/playable-entities) to demonstrate the built-in actions. These interactions enable us to simulate complex behaviors and tasks in open-world environments, showing its potential to support a wide range of embodied tasks in future research. Furthermore, Unreal Engine's flexibility allows us to customize and add new actions as community needs arise.

---

> ### Author Response · Authors · 2024-11-26
> **Response to Reviewer u5UD (2)**
>
> >**W5:** There are no language-modal input/communication among agents. The multi-agent feature is not highlighted.
>
> **R5**: Language-based input and agent communication can be implemented through the gym interface without modifying the UE environment. Researchers can easily create custom multi-agent interaction tasks using this interface, e.g., cooperative navigation with communication. The primary challenge in multi-agent scenarios stems from the increasing computational and interaction costs that arise as more agents are added. As shown in Table 2, UnrealZoo demonstrates improved efficiency in supporting multi-agent interactions.
>
> >**Q1:** About the choice of base engine, is UE the only engine that can render photo-realistic images?
>
> **R6**: While Unreal Engine (UE) is not the only engine capable of rendering photorealistic images—others include Omniverse, Unity, and Blender—UE's primary advantage lies in its open-source and mature ecosystem. Its open-source nature allows us to modify the game engine for potential future features. The Epic Marketplace/Fab provides a wide range of high-quality content created by professional developers, allowing researchers and environment developers to access diverse assets and tools that accelerate development and enhance realism in simulated environments. This enables us to scale up our collection of environments with minimal effort. Additionally, numerous developers, particularly those working on AAA video games, actively contribute to Unreal Engine, ensuring we benefit from continuous updates to the game engine, plugins, and content.
>
> >**Q2:** There is a lot of Agent body mentioned in Table 1. Are they driven by pre-defined policy? What are their action spaces?
>
> **R7:** Except for the drone, all agents are controlled using basic movement parameters of angular velocity and linear velocity, which can be adjusted either step-wise by the user or automatically through the built-in navigation system. The drone, on the other hand, is controlled by applying forces along the x, y, and z axes, as well as yaw rotation, offering fine-grained movement capabilities. Special interactive actions, such as opening doors, picking up objects, or simulating falls, are executed through discrete control signals. We provided detailed usage APIs and video demonstrations on this [website](https://unrealzoo.notion.site/playable-entities), offering further insights into the agents' control mechanisms and capabilities.
>
> >**Q3:** Cannot find Table 3.3 in line 308 and Table 4.1 in line 368. GPT4-o in Table 5 should be GPT-4o.
>
> **R8**: We apologize for the confusion. We have corrected the table references in the paper. The correct references are now Table 2 (line 308) and Table 3 (line 368). We have also changed GPT4-o to GPT-4o. We appreciate your careful review, which has helped us improve the clarity of our paper.
>
> >**Q4:** Why GPT-4o performs so poorly? Do you do the error breakdown analysis? Can you use other information like position or third-person view to increase the success rate?
>
> **R9:** Thanks for the suggestion. To provide clarity, we have recorded videos showing how the GPT-4o agent working on both navigation and tracking tasks. The videos can be found [here](https://unrealzoo.notion.site/vlm-agent?pvs=4). Roughly speaking, we find that GPT-4o's lack of 3D spatial reasoning with ego-centric observation and the heavy latency in inference will also make the GPT-4o agent fail in dynamic environments, e.g., tracking.
>
> To ensure a fair comparison, we evaluated all agents using identical observation space in each task. For active tracking, agents received only first-person images, while navigation tasks provided both first-person images and the target's relative coordinates.
>
> >**Q5:** In the social tracking experiment, as the training set grows, the growth of SR is intuitive. Can you conduct the experiment on the same scale training (compared to the less diverse data) to give a fair evaluation?
>
> **R10**: We trained the offline RL policy using the same amount of training data. Each of the three datasets (1 Envs., 2 Envs., 8 Envs.) contains exactly $100k$ samples, ensuring a fair comparison. The details of collected data are introduced in Appendix `B.4`.

---

> > ### Comment · Reviewer_u5UD · 2024-11-29
> >
> > Thanks for the detailed responses.
> > I have more questions.
> >
> > - Can you give a more detailed analysis of simulation speed?
> > - I have seen that this platform is feasible in training an agent that moves in the field. However, I am concerned about the physical simulation capabilities of the platform. Based on the author's video demo, it seems that object interactions (opening doors, picking up objects), etc. are now simulated through some APIs without actual physical processes. Therefore I am not sure if the platform can train an embodied agent that can be migrated to a real environment.
> > - I didn't see  GPT failure case on visual nav task in provided videos.   I don't agree with GPT-4o's lack of 3D spatial reasoning. In works such as [1], LLM-based spatial reasoning can perform very well.
> >
> > [1] Fang, Jiading, et al. "Transcrib3D: 3D Referring Expression Resolution through Large Language Models." arXiv preprint arXiv:2404.19221 (2024).

---

> > > ### Author Response · Authors · 2024-12-01
> > > **Response to Reviewer u5UD (3)**
> > >
> > > Thanks for your valuable feedback. The following is our response to your new questions:
> > >
> > > > Can you give a more detailed analysis of simulation speed?
> > >
> > > **R1:** We are sorry for not introducing the details in the paper. The simulation speed is evaluated on an Nvidia GTX 4090 GPU, Intel i7-14700k CPU, and Windows OS. We repeatedly call each Python function (`img = unrealcv.get_image(image_type)`) 1000 times and record the time $T$ it takes. The multi-agent interaction is evaluated by calling the gym interface (`obs = env.step(actions)`). Then we calculate $FPS = 1000/T$. The image resolution is 640x480. Here we report the detailed results on environments at different scales in the following table:
> > >
> > > | **Env** | **Color Image** | **Object Mask** | **Surface Normal** | **Depth Image** | **2 Agents Interaction** | **6 Agents Interaction** | **10 Agents Interaction** |
> > > |---|---|---|---|---|---|---|---|
> > > | [FlexibleRoom](https://unrealzoo.notion.site/FlexibleRoom-beecf355e71a49c6a8467d5e6decef6a) (71 objects, 2440m^2^)  | 85 | 164 | 137 | 100 | 94 |40 |27|
> > > | [BrassGarden](https://unrealzoo.notion.site/City-of-Brass-Gardens-a0cada04381f4458b7be7c5456945c61) (467 objects, 9900m^2^) | 107 | 214 | 173 | 123 |  102 |48|31|
> > > | [Supermarket](https://unrealzoo.notion.site/Supermarket-ec19650537344142927d60617ff4cf74) (2839 objects, 11700m^2^)  | 99 | 173 | 167 | 117 | 70 |33|19|
> > > | [SuburbNeighborhoodDay](https://unrealzoo.notion.site/SuburbNeighborhood_Day-4eab6005bf1e46418e1860eb6c85897f) (2469 objects, 23100m^2^)  | 79 | 139 | 112 | 92 | 53 |27|17|
> > > | [GreekIsland](https://unrealzoo.notion.site/Greek-Island-beecf355e71a49c6a8467d5e6decef6a) (3174 objects, 448800m^2^) | 82 | 167 | 124 |103  |  52 |23|16|
> > > | [MedievalNatureEnvironment](https://unrealzoo.notion.site/Medieval-Nature-Environment-0706190c7b2641a98de714bf546f7e0a) (8534 objects, 16km^2^) | 70 | 112 | 97|74|29|14|10|
> > >
> > > **We will add these details in the final version.**
> > >
> > > >I am concerned about the physical simulation capabilities of the platform.... I am not sure if the platform can train an embodied agent that can be migrated to a real environment.
> > >
> > > **R2:** We understand your concern. We would like to emphasize that we are not the first work that employs Unreal Engine to build virtual worlds for embodied AI. Some previous works have demonstrated that the models trained in Unreal Engine can be deployed on various real-world scenes and robots, including drones [1, 2], wheeled robots[3,4], and quadruped robots [5]. We also provide the environment augmentation wrapper in the gym toolkits to help the users automatically randomize the environment, including the layouts, textures, sizes, illumination, and even the hyperparameters of the actions (to enrich the animated interactions), to enhance the sim2real generalization by diversifying the data.
> > >
> > > Beondy sim2real transferability, there are several fundamental challenges in embodied AI worth exploring, including multi-modal reasoning[6], planning in dynamic environments[7], and social interactions[8,9,10]. **Physics-based manipulation is not essential for studying these advanced topics in embodied AI.** Therefore, our current focus is not on developing low-level controllers for object manipulation using high-fidelity physics simulations. Instead, we assume that the physics-based object manipulation skills have been developed, and utilize animation-level control APIs to streamline these actions, a method also employed in simulators like VirtualHome and LEGENT. Numerous studies[6-10] have shown that this approach allows us to bypass the limitations of low-level controllers, enabling the exploration of emerging topics in embodied AI.
> > >
> > > Additionally, Unreal Engine supports physics-based motion control, allowing us to extend our playable entities and APIs in the future for high-fidelity simulations of agent locomotion and interactions with objects.
> > >
> > >
> > > References:
> > >
> > > [1]"Enhancing optical-flow-based control by learning visual appearance cues for flying robots." Nature Machine Intelligence
> > >
> > > [2]"Air-M: A Visual Reality Many-Agent Reinforcement Learning Platform for Large-Scale Aerial Unmanned System." IROS 2023.
> > >
> > > [3]"End-to-end Active Object Tracking and Its Real-world Deployment via Reinforcement Learning." IEEE T-PAMI, 2019.
> > >
> > > [4]"Empowering Embodied Visual Tracking with Visual Foundation Models and Offline RL." ECCV 2024.
> > >
> > > [5]"RSPT: Reconstruct Surroundings and Predict Trajectories for Generalizable Active Object Tracking." AAAI 2023.
> > >
> > > [6]"MMToM-QA: Multimodal Theory of Mind Question Answering." ACL 2024 (Outstanding Paper Award).
> > >
> > > [7]"HAZARD Challenge: Embodied Decision Making in Dynamically Changing Environments." ICLR 2024.
> > >
> > > [8]"NOPA: Neurally-guided Online Probabilistic Assistance for Building Socially Intelligent Home Assistants." ICRA 2023.
> > >
> > > [9]"Watch-And-Help: A Challenge for Social Perception and Human-AI Collaboration." ICLR 2021 (Spotlight).
> > >
> > > [10]"Building Cooperative Embodied Agents Modularly with LLMs." ICLR 2024.

---

> ### Author Response · Authors · 2024-12-01
> **Response to Reviewer u5UD (4)**
>
> > I didn't see GPT failure case on visual nav task in provided videos. I don't agree with GPT-4o's lack of 3D spatial reasoning. In works such as [1], LLM-based spatial reasoning can perform very well.
>
> **R3:**  We apologize for not providing the failure case earlier. Now, the failure case analysis has been updated on this [website](https://unrealzoo.notion.site/vlm-agent). The initial showcase is to demonstrate that GPT-4o is still very inefficient in planning and frequently takes some additional or wrong actions, even though it can sometimes complete this task. However, in our experiments, the number of failure cases far exceeds the number of successful ones. We have added the normal failure cases with analysis on our website to provide more context.
>
> For 3D spatial reasoning, we utilize GPT-4o, a **large vision language model (VLM)**, to process `raw pixel 2D images` and task descriptions as input and produce actions in an end-to-end manner without additional computer vision models. In contrast, Transcrib3D employs an LLM-based method that incorporates an additional **3D object detector** to process `colored point clouds` and generate a comprehensive list of objects as text for **large language models (LLMs)**. Therefore, this work can not demonstrate the 3D spatial reasoning capabilities of VLMs, as it relies on LLMs instead.
>
> To clarify, we do not deny that GPT-4o is of fundamental multimodal reasoning abilities, such as identifying target categories in images and providing detailed scene descriptions. However, based on our own experiments and other recent works([Chen, et al. CVPR 2024](https://openaccess.thecvf.com/content/CVPR2024/papers/Chen_SpatialVLM_Endowing_Vision-Language_Models_with_Spatial_Reasoning_Capabilities_CVPR_2024_paper.pdf) and [Feng, et al. SIGRAPH Asia](https://arxiv.org/abs/2410.16991)), it is clear that the **3D spatial reasoning ability of VLMs remains limited**, e.g., GPT-4o or GPT-4V. They often generate **hallucinated responses**. Our newly updated failure demo on the [website](https://unrealzoo.notion.site/vlm-agent) also highlights such problems. For instance, in a key moment where a left turn was needed, GPT mistakenly suggested a right turn. This led to a series of cumulative errors and ultimately resulted in the failure to accomplish the task. As validated by Chen, et al., **this limitation is primarily due to the training datasets used, which typically lack spatially rich data or high-quality human annotations for 3D-aware queries**, rather than the model's inherent architecture. Inspired by this work, a future application of our environments could automatically generate large-scale images and videos with detailed 3D annotations to improve the spatial reasoning capabilities of VLMs. We will include these clarifications in the revision.

---

> > ### Comment · Reviewer_u5UD · 2024-12-01
> >
> > Thanks. I don't have any more questions at the moment.

---

> > > ### Author Response · Authors · 2024-12-01
> > > **Thanks for your postive feedbacks**
> > >
> > > We would like to extend our appreciation for your positive feedback and score improvement.  We’re glad our responses addressed your concerns. Once again, thank you for your time and your thoughtful review. Your kindness and insights will encourage us to continuously contribute to the field.

---

### Official Review · Reviewer_dbRo · 2024-10-30

**Soundness:** 2
**Presentation:** 2
**Contribution:** 3
**Rating:** 3
**Confidence:** 4

**Summary:**

This paper introduces UnrealZoo, a collection of photorealistic environments based on the Unreal game engine. These environments serve as a platform for training and analyzing embodied agents in highly realistic virtual environments. Moreover, unlike previous works that limit to small scenarios (e.g., a kitchen or a room), UnrealZoo provides vast virtual environments with carefully designed assets that resemble the challenges that agents operating in the real world would face. UnrealZoo is based on UnrealCV, which authors have modified to suit the needs of embodied agent research, improving its usability and performance (FPS). Experiments demonstrate the usability of UnrealZoo for embodied agent research, showcasing applications such as visual navigation, social tracking, and more. Results highlight the importance of diversity in the training environments for offline RL methods, how these fail to generalize across embodiments, and show the limitations of VLM models in highly complex realistic environments.

**Strengths:**

- The paper introduces realistic, highly complex open-world scenarios that resemble real-life scenarios embodied agents would face in the real world. UnrealZoo fills the gap between open-world environments that are far from the complexity of real-life scenarios (e.g., MineDojo, NetHack Learning Environment, or Craftax) and realistic environments that are usually far from the vast scenarios typically encountered in real life (e.g.,
ThreeDWorld or Habitat 3).

- UnrealZoo provides 100 scenes of very different natures and many playable entities (humans, animals, cars, robots, etc.). Although realistic virtual environments exist for embodied AI research, these often focus on some specific domain (e.g., CARLA and autonomous driving). The variety of the UnrealZoo environments makes the proposed framework a very versatile tool for embodied AI research.

- Although tools exist for creating environments in the Unreal game engine and libraries for computer vision (UnrealCV), the authors modify UnrealCV to suit the needs of embodied agent research. Moreover, they make some modifications to the agent-environment communication and the rendering pipeline improving the performance (frames per second) of the environments significantly.  Finally, the authors provide an easy-to-use but versatile API based on OpenAI Gym.

**Weaknesses:**

Although I think that the main contribution of this work (UnrealZoo) could be of great relevance to the embodied AI field, I have two major concerns that I strongly believe should be addressed before:

**Concern 1:**  In lines 185-187 authors state that the environments are sourced (and paid) from the Unreal Engine Marketplace (now renamed as Fab, see https://www.fab.com/), and in lines 236-238 they mention: *"[...] we will package the projects and release binaries for the community."*. However, the standard license of Fab states that (verbatim from section 5.a of https://www.fab.com/eula): *"Under a Standard License, you may not Distribute Content on a standalone basis to third parties [...]"*. Please **make sure** that distributing the binaries as mentioned in the paper is legal under the terms in which the assets were purchased.

Moreover, if the binary distribution of the environments is legal, the fact that this has to be shipped in a binary package greatly limits the open-source nature of the project. For example, if a contributor or developer wants to modify existing UnrealZoo environments, they would be greatly limited by the binary format of the Unreal environment. I'm open to discussion and willing to read the responses of the authors in this regard.

**Concern 2:**  I strongly believe that the overall presentation and writing quality of the paper should be improved. Examples:

- Lines 123 to 142 discuss previous literature on virtual environments. This text uses the "virtual environments" term which is very generic, but only discusses environments based on realistic simulators. Please modify the text to explicitly refer to **only** realistic simulator-based environments, or include the vast literature on environments that don't simulate the real world (e.g., NetHack, MineDojo, MineRL, Craftax, Atari, VizDoom,...).

- Missing details in the experimentation (Section 4). The authors do not provide any detail (or reference) on the specific methods used for the experiments. For example, the online and offline RL methods are unknown, there is no mention of the NN architectures employed. Moreover, encourage the authors to include the training curves of the methods (at least in the appendix). Currently, there is no evidence of the level of convergence of the methods. Furthermore, there is no information on how the human benchmark was collected: how many humans have been used? Were the humans informed on the task to solve or only had access to reward values? Were the humans experienced in video games?

- In L402 authors define the "Average Accumulated Reward" metric. I believe the authors refer to the average episodic return typically employed in RL. If this is the case, I think that common terms are much preferable instead of introducing new terminology.

- Section in the appendix should be listed with letters, not with numbers. For example, the first section of the appendix should not be "7 Data", but "A Data". I believe this could be fixed by including the "\appendix" command in LaTeX. Moreover, the appendix sections include many of the minor issues from the main text (some are pointed out below). Please consider also proofreading the appendix sections.

- The references section should be carefully checked and improved. These are some of the issues, but please check all the references:
    + Some papers referenced as preprints have been published in conferences or journals. For example, the CARLA paper was published in CoRL 2017 but is listed as an arXiv preprint.
    + Incorrect capitalization in many titles. Example: CARLA in Dosovitskiy et al. 2017 should be all uppercase.
    + Some names have been abbreviated, for example: in Gaidon et al. 2016. The same reference is missing the full name of the conference.
    + Gupta et al. 2017 and many other references have inconsistent use of capitalization.
    +  Some references include the acronym of the conference and others don't.
    + Inconsistent format of the proceedings name, for example: some include the year, some don't.
    + Some references are included as @article when should be @inproceedings. Example: Yuan et al. 2020.

**Minor issues:**

I found many minor writing and presentation issues in the paper, in the following I list some of them. The issues are individually minor, but many minor minor issues add a major issue. Please consider an exhaustive revision of the full paper.

- When listing the contributions (last part of the intro) the numbers jump from 1) to 3), missing 2).
- The caption of Table 1 is missing the final dot. Moreover, the employed icons should be explained in detail somewhere in the main text or the appendix.
-  L197: "[...]Project Website [...]" should be lowercase.
- L309: "To be user-friendly [...]"; L311: "In this way, the beginner can easily use and customize [...]", reformulate to maintain formalism.
- L349: "visual Navigation tasks [...]" fix capitalization.
- L350 and L351 have missing whitespace before and after parenthesis.
- L368: "In Table. 4.1". Extra dot after Table and there's no Table 4.1, I believe the authors refer to Table 3.
- Table 3 does not define what the numbers on the table represent. I believe these are EL and SR, but these should be explicitly mentioned.
- L413: missing whitespace before parenthesis.
- L414: extra dot inserted after "Figure".
- L428: "[...] are provided in the appendix.",  please specify the appendix section.
- L431: extra dot and whitespace inserted after "Figure".

I want to emphasize that I believe that the contribution of the work is relevant to the field, thus, I'm willing to update my score if the authors address the mentioned concerns and issues.

**Questions:**

- **Q1:** The paper includes experiments on the cross-embodiment generalization capabilities of some offline RL, however, it does not motivate cross-embodiment generalization per se, which I find non-trivial. What is the motivation behind cross-embodiment? Why is cross-embodiment generalization an interesting capability for embodied agents? Is there any practical or real-life scenario where cross-embodiment generalization is relevant?

- **Q2:** What happens when the agent's implementation can not handle the control frequency specified by the environment? Are the actions repeated or no action is taken (something like no-op in some RL environments)? Is it possible to freeze the environment to wait for the action of the agent?

---

> ### Author Response · Authors · 2024-11-26
> **Response to Reviewer dbRo (1)**
>
> Thank you for your valuable feedback. Below, we address each of your points in detail.
>
> >**W1-1:** License issue.
>
> **R1**: We **do not violate the mentioned statements (section 5.a in Fab)** in eula, as we do not share the content on a **standalone basis (in source format)**. Instead, we package all the content in an executive binary project, similar to the binaries in [unrealcv model zoo](https://docs.unrealcv.org/en/latest/reference/model_zoo.html). Distributing such executive binary is allowed by [Epic Content License Agreement 3.a ](https://www.unrealengine.com/en-US/eula/content):
>
> *` You may Distribute Licensed Content incorporated in object code format only as an inseparable part of a Project to end users.…...This means, for example, you may Distribute software applications (such as video games) that include Licensed Content to the general public, whether directly by you or through a distributor or publisher.`*
>
> Note that all the contents in UnrealZoo were purchased from Unreal Engine Marketplace before Oct.1 (When Epic Marketplace was replaced by Fab), so we refer to the Epic Content License instead of Fab. Besides, the binary distribution is also allowed by Fab License(https://www.fab.com/eula), referring to section 4 rather than section 5. This enables us to further purchase the environments in Fab to scale up the collection in the future.
>
> >**W1-2:**  Moreover, if the binary distribution of the environments is legal, the fact that this has to be shipped in a binary package greatly limits the open-source nature of the project.
>
> **R2**: The [UnrealCV command system](https://docs.unrealcv.org/en/latest/reference/commands.html) provides a set of built-in commands to support various run-time modifications, enabling users to flexiblely manipulate content in the packaged environment. Moreover, in UnrealZoo, we further add a set of blueprint functions, which is callable by UnrealCV, to support more advanced environment customization, such as changing the textures, lighting, or weather.
>
> Following UnrealCV, the UnrealCV+ plugin also uses the MIT License. We will open-source UnrealCV+ to help contributor package their own projects and share them with the community.
>
> There is a trade-off between flexibility and ease of use. Although UnrealCV has been available for 8 years, most AI researchers tend to prefer using off-the-shelf binaries rather than learning Unreal Engine Editor and creating new environments from scratch. We believe that providing a large-scale collection of pre-built environments and playable entities for the AI community will help researchers bypass the costly process of building virtual worlds and explore new avenues in their field.
>
> >**W2:** I strongly believe that the overall presentation and writing quality of the paper should be improved.
>
> **R3**: Thank you for your detailed comments. Following your suggestions, we have addressed all the listed issues in the revision. The main updates include:
>
> - Revised the text in the related work section to focus on realistic simulators and added two recent works for comparison.
> - Added the implementation details in Appendix `C`, covering RL algorithms, NN architectures, hyperparameters, human benchmarks, and prompts for VLM agents.
> - Added the results section in Appendix `D` with training curves and detailed results for each environment.
> - Replaced "accumulated reward" with "episodic return" in the evaluation metrics.
> - Restructured the appendices:
>     - A UE Environment
>     - B Exemplar Tasks
>     - C Implementation Details of Agents
>     - D Additional Results
> - Included a table in Appendix `A` to clarify the icons used in Table 1 and a figure comparing the visual realism of different engines.
> - Added snapshots of all environments used in the experiments to the Appendix.
> - Revised the reference formatting as per your suggestions.
> - Fixed all minor issues.

---

> ### Author Response · Authors · 2024-11-26
> **Response to Reviewer dbRo (2)**
>
> >**Q1:** What is the motivation behind **cross-embodiment**? Why is cross-embodiment generalization an interesting capability for embodied agents? Is there any practical or real-life scenario where cross-embodiment generalization is relevant?
>
> **R4**:  In the real world, robots typically have different mechanical structures and hardware configurations. Training separate policies for each hardware configuration is inefficient. Therefore, we need to build generalizable agents that can work across different embodiments. For example, if a tracking agent has good cross-embodiment capabilities, we can efficiently deploy one model across quadruped robots, wheeled robots, or even humanoid robots, significantly reducing the costs.
>
> >**Q2:**  What happens when the agent's implementation can not handle the control frequency specified by the environment? Are the actions repeated or no action is taken (something like no-op in some RL environments)? Is it possible to freeze the environment to wait for the action of the agent?
>
> **R5**: Without frequency control, the agent runs at 6~8 FPS, as the vision fondation model taks about 150ms at each step. Note that the results in Figure 5 and Table 5 are evaluated without frequency control. We add the new results (w/o control) in Table 4 for reference.
>
> The frequency control is implemented by adjusting the simulator's clock based on the interaction frequency between the agent and the environment. We simulate real-world closed-loop control frequency by controlling the ratio between simulated time and engine time. This approach eliminates the need for repeated actions or no-op states to control frequency. Compared to repeated control, time dilation can maximize computational resource utilization instead of wasting time repeating actions.
>
> You can also freeze the environment to wait for the agent's action by calling the unrealcv command `vset /action/game/pause` in Python.

---

> > ### Comment · Reviewer_dbRo · 2024-11-28
> >
> > First of all, thanks for answering the questions and weaknesses raised.
> >
> > Regarding the first weakness, I'm happy to see that there is no license issue if assets are shared in their binary format. However, I continue to believe that the closed-source and binary-only nature of some parts of the project greatly limits third-party contributions or modifying existing environments. Note that, in my opinion, this is just a minor concern that does not affect the contribution of this work to the field.
> >
> > Although my first concern has been solved, the second concern (presentation and writing, which I think is very important) has been just partially addressed. I believe the changes introduced to the paper have improved the original version, however, a significant workload is still required. I have revised the new version of the paper, and I see that the authors have corrected many of the issues I pointed out in the last message. However, as mentioned, these issues were only **some examples**, and I think that the paper continues to have a significant margin for improvement regarding writing and presentation. To point a few:
> >
> > - Table 5 does not describe what the shown numbers are. Again, I can sense what they mean, but tables must describe what they display, at least what the numbers mean.
> >
> > - I appreciate the work done to include the new appendix sections, however, the appendix sections must be referenced in the main text. I only see appendix sections B1, C1, and C2 in the main body of the paper. For example, Table 1 should point to Appendix A.1, but this is not the case.
> >
> > - Thanks for including the descriptions of each emoji, however, Table 6 in Appendix A.1 **is not a real table, is an image** (and somewhat blurry). Please use a proper LaTeX table.
> >
> > - The references section continues to have many issues and inconsistencies. For instance, some use *"In Proceedings"*, others just employ *"In"*; There are many incorrect (or missing) use of capitalization (e.g., Kolve et al. 2017, AI2-THOR); some references continue to be `@article` instead of `@inproceedings`.
> >
> > Overall, I continue to believe that the paper requires a significant amount of work before being published. In my opinion, the required work is much more than the one that should be involved in preparing a camera-ready version of the paper, which makes me hold my original score. I encourage the authors to continue working on the paper, as it can be a great contribution to the field.

---

> ### Author Response · Authors · 2024-11-28
> **We sincerely hope you can re-evaluate your rating**
>
> Thank you for your thorough review and constructive feedback on our manuscript. We appreciate the time and effort you have invested in evaluating our work and are committed to addressing the concerns you have raised.
>
> 1. We emphasize that the source link for each environment is available in [scene gallery](https://unrealzoo.notion.site/scene-gallery). Hence, anyone needing to modify the environment from the source can directly buy and download the licensed content from the public marketplace. With the licensed content, the following developers only need to apply our open-sourced UnrealCV+ Plugins in the UE project to reproduce the environment in UnrealZoo. There is no free lunch! **If the community only pursues completely free and open-source content, artists will be discouraged from contributing more high-quality content and hinder the researchers from accessing large-scale high-quality environments, ultimately slowing the advancement of the related AI fields.** Therefore, we believe our distribution strategy effectively balances the trade-off between diversity of content and customization flexibility for contributors. In your initial reviews, you mentioned that the license and open-source issue is one of your two main concerns. Please be consistent with your initial comment and reevaluate your rating.
>
> 2. We agree that the writing of a paper should be standardized and rigorous, and we will continue to iterate and improve the writing of the paper for this goal. However, most of the issues you raised have been fixed or can be fixed in the final version. **So we argue that these issues in the presentation should not be the main reason for rejecting this paper.** The issues you recently mentioned are very minor, and they can be fixed within a few minutes. In our current version, we have:
>  - describe the meanings of the number in the Caption of Table 5, i.e.,*"Each cell presents three metrics from left to right: Average Episodic Return (ER), Average Episode Length (EL), and Success Rate (SR)."*
> - In the current version, we directly refer to the related Table and Figure included in the Appendix. For example, in the caption of Table 1 (L109-L110), we mentioned `Table 6` and `Figure 6` (Appendix A) for convenience. In L 415, we mentioned Figure 10 (Appendix A) and Figure 14 (Appendix D). In L431, we mentioned Table 11 (Appendix D). If you prefer references to the corresponding sections in the Appendix, we will include them in the final version.
> -  We apologize for the oversight with Table 6 and have replaced the image with a proper LaTeX table, which is now clear and fits the academic standards of the paper.
> - We apologize for the oversight of the reference issues. Note that not all the conferences have proceedings, e.g., ICLR. So it is reasonable that some conferences do not have *"Proceedings"*. AI2-THOR has been updated, and we will further fix the other misuse capitalization issues in the final version. The journal and Arxiv papers use the `@article`, and the conference papers use `@proceedings`.
>
> Overall, we appreciate your detailed comments to improve our writing. Thank you again for your valuable feedback. We will continuously work on this work to improve the presentation and make our contribution as valuable as possible to the community.
> **We sincerely hope you will re-evaluate this work and provide a fair and objective score.**

---

> > ### Comment · Reviewer_dbRo · 2024-11-29
> >
> > Again, the issues I listed were **only a few examples** of many writing issues I found in the paper. Overall, the issues sum up, creating a problem that I believe **is not minor**. Moreover, I cannot review a new version of the paper which can not be uploaded in this period of the rebuttal, thus, I have to make decisions on the last version that the authors uploaded, which in my opinion, is not marginally below the acceptance threshold. I strongly believe the paper needs some rewriting work that is not a matter of *"a few minutes"*, as the authors mentioned.

---

> > > ### Author Response · Authors · 2024-12-01
> > > **We sincerely hope you can reconsider your evaluation and give a fair and objective score.**
> > >
> > > Thanks for your feedback. But we do not agree with your point. The following is our reasonings:
> > > - **First, some of the listed issues in your recent feedback are not issues at all.** For example, we have directly referenced most Tables and Figures in the Appendix for convenience, rather than citing the related sections, which we consider **a matter of writing preference rather than an issue.** Regarding the references, after referring to numerous accepted papers at ICLR 2024, including outstanding papers, we have not found any that meet your standards. Could you provide an example from accepted ICLR papers to clarify your criteria? Referring to these accepted papers, rather than your preference, we are also confused about **how many cases you listed are real issues**.
> > > - **Second, it truly took us *a few minutes* to confirm and address the examplar issues you recently listed.** To be specific, you posted your comments about one hour before the closing of the paper submission system. Once we noticed them (about 30 minutes later), we managed to fix most of the cases with just a few minutes remaining and successfully updated the revision on OpenReview. We believe that the issues you raised are likely more significant than those not mentioned, suggesting that other potential issues can also be addressed in "a few minutes". We acknowledge that thorough proofreading is still necessary to enhance the presentation of the final version. For example, we have noticed the issues in Table 2, and we will fix the typos (Suf.->Sur. Nor. and NorDepth->Depth) and add more details in the caption in the revision.
> > > - **Third, it is `unfair`, `subjective`, and `unprofessional` to evaluate the value of a work based on these minor writing issues (some even may not be), disregarding its significant contributions.** For a top-tier conference, it is natural to require high standards in writing, but **high standards do not mean nitpicking**. Instead, the contribution of a work to its related domains and community should be prioritized in the evaluation. Referring to your comments ( *"The main contribution of this work (UnrealZoo) could be of great relevance to the embodied AI field"*, *"It can be a great contribution to the field."*), we firmly believe your rating is `biased`.
> > >
> > > **Once again, we sincerely hope you can reconsider your evaluation and give a fair and objective score.**

---

### Official Review · Reviewer_BMxe · 2024-11-03

**Soundness:** 3
**Presentation:** 3
**Contribution:** 2
**Rating:** 5
**Confidence:** 4

**Summary:**

This paper proposed UnrealZoo, a photorealistic and large-scale environment for embodied AI agents. In UnrealZoo, agents can perform much more complicated actions than just traditional navigation, such as jumping and climbing. Further, agents can control vehicles, humanoids, animals, etc, allowing experimentation with different embodiments.

The authors propose and instantiate various tasks in their proposed simulator. They evaluate both VLMs and RL-trained agents on a subset of their proposed tasks.

**Strengths:**

The visuals from the proposed simulator look great.

There are a variety of task and environments. Environments also have a wide variety of scales.

The authors improved the performance of the rendering pipeline.

The environment supports multiple agents.

**Weaknesses:**

My biggest concern with this paper is that the proposed tasks and simulator lack a defined direction.

There has been a considerable amount of interest centered around indoor navigation with the goal of sim2real transfer. Indoor navigation has been the subject of focus due to real robots that can serve as a deployment target existing, albeit it far from perfect hardware. The reviewer is unaware of any existing hardware platforms that would be sensible deployment targets.

There is also considerable interest in environments to evaluate new reinforcement learning methods and algorithms. While there are multiple criteria for these environments, one key one is speed -- the environment itself must be very performant as the ability to quickly iterate on new ideas is key. UnrealZoo is unfortunately very slow by these standards.

While I do find the proposed simulator, tasks, and environments interesting, I am concerned about the value of the contribution.

**Questions:**

N/A

---

> ### Author Response · Authors · 2024-11-26
> **Response to Reviewer BMxe (1)**
>
> Thank you for your valuable feedback. Below, we address each of your points in detail.
>
> >**W1:** My biggest concern with this paper is that the proposed tasks and simulator lack a defined direction. The reviewer is unaware of any existing hardware platforms that would be sensible deployment targets.
>
> **R1:**  Our ultimate goal is to simulate complex open worlds to build generalist embodied agents for real-world applications. The main limitation of previous simulators is their **lack of diversity in scenes and characters**. Many focus on specific scenarios, such as daily home activities or urban autonomous driving, which hinders the advancement of generalist embodied AI in open worlds. UnrealZoo provides various playable entities, including humans, quadcopters, quadruped robots, cars, animals, and motorcycles. Several related works[1-5] have demonstrated the feasibility of transferring the models trained in Unreal Engine to robotic hardware platforms, including drones [1, 2], wheeled robots[3,4], and quadruped robots [5]. UnrealZoo can be used for generating high-quality ego-centric data in complex scenes, speeding up the training process, and evaluating the robustness of the learned policy before real-world deployment, showing the potential applications of UnrealZoo for robotic hardware deployment.
>
> Currently, our primary focus is exploring fundamental challenges in embodied perception, reasoning, decision-making, and learning, with an emphasis on **adaptability** and **generalization** in **scaling complex virtual worlds**. Deploying the agents trained in UnrealZoo across different existing hardware platforms, such as drones, humanoid robots, and quadruped robots, in complex real-world scenarios will be our future work.
>
> References:
>
> [1]"Enhancing optical-flow-based control by learning visual appearance cues for flying robots." Nature Machine Intelligence
>
> [2]"Air-M: A Visual Reality Many-Agent Reinforcement Learning Platform for Large-Scale Aerial Unmanned System." IROS 2023.
>
> [3]"End-to-end Active Object Tracking and Its Real-world Deployment via Reinforcement Learning." IEEE T-PAMI, 2019.
>
> [4]"Empowering Embodied Visual Tracking with Visual Foundation Models and Offline RL." ECCV 2024.
>
> [5]"RSPT: Reconstruct Surroundings and Predict Trajectories for Generalizable Active Object Tracking." AAAI 2023.
>
> >**W2:**  While there are multiple criteria for these environments, one key one is **speed** -- the environment itself must be very performant as the ability to quickly iterate on new ideas is key. UnrealZoo is unfortunately very slow by these standards.
>
> **R2:** We acknowledge the importance of speed as a criterion for evaluating simulation environments. UnrealZoo achieved comparable performance to other photorealistic environments, such as ThreeDWorld (based on Unity) and OmniGibson (based on Omniverse). Our experiments have demonstrated that the RL agents can be trained in UnrealZoo with an affordable time cost (1~6 hours) using a customer-level GPU (RTX 4090).
>
> Fast prototyping and iteration can be achieved through various methods that do not only depend on high interaction frequencies within the environment. Techniques such as distributed training, parallel execution across multiple virtual environments, and offline reinforcement learning effectively address the interaction speed limitations of online RL. The advent of high-fidelity simulation environments has also made it possible for researchers to collect large volumes of high-quality data. Offline RL and imitation learning, in particular, leverage these pre-collected datasets for efficient policy learning, significantly reducing reliance on real-time interactions during training.  We believe that as these learning approaches gain traction, high-fidelity environments like UnrealZoo will play a crucial role in supporting the development and evaluation of embodied agents.

---

> ### Author Response · Authors · 2024-11-26
> **Response to Reviewer BMxe (2)**
>
> > **W3:** I am concerned about the value of the contribution.
>
> **R3**: Collecting diverse photorealistic scenes at scale can simulate various real-world challenges that occur in the open world. essential for training robust agents and evaluating their generalization for real-world deployment. Note that UnrealCV has been broadly applied in a great number of pioneer works in visual intelligence for data generation, model diagnosis, and interactions, as shown in [the paper list](https://github.com/unrealcv/papers-with-unrealcv). As UnreaCV+ introduced in UnrealZoo is compatible with the UnrealCV command system, the researchers in the related areas, such as semantic understanding, 3D vision, and embodied vision, can directly access UnrealZoo and benefit from the enriched photorealistic environments, e.g., scaling data generation, diagnosis the vision models in more complex environments, or training more robust interactive policy.
>
> Besides, with the advancement in large vision-language models (VLM), such photorealistic virtual worlds with unprecedented complexity and diversity are demanded by the community to further benchmark the VLM agents, particularly for scene understanding, dynamic modeling, social reasoning, and long-horizon decision-making.
>
> Moreover, the toolkit (the UnrealCV+, gym interface, and wrappers) we developed for UnrealZoo can mitigate the domain gap between the game developer and the AI researchers, beneficial to both communities. For example, game developers can use these tools to effectively build AI agents as NPCs in their games, and AI researchers can more conveniently connect to the AAA game to collect high-fidelity data and benchmark the models/algorithms.

---

> ### Author Response · Authors · 2024-12-02
>
> We sincerely thank you for your thoughtful review. **As the discussion period draws to a close, we would appreciate your feedback to confirm whether our replies have addressed your concerns.** If you have any remaining questions, we are happy to provide further clarification. If our responses have resolved your concerns, we would be deeply grateful if you could consider raising the score. Thank you again for your time and effort during the review process.

---

### Official Review · Reviewer_7kEB · 2024-11-04

**Soundness:** 3
**Presentation:** 3
**Contribution:** 3
**Rating:** 6
**Confidence:** 4

**Summary:**

This paper introduces UnrealZoo, a collection of 100 3D environments based on top of the UnrealCV engine. This collection of environments is novel since it spans a variety of scales, agent types, has a rich navigation system and is multi agent.

**Strengths:**

Originality: The paper demonstrates originality by contributing large scale diverse environments that can't be found for training RL agents today. Performance of RL agents on such environments from different playable entities hasn't been studied in detail before and having such environments would allow for that.
Quality: Inclusion of images, algorithmic benchmarks, and comparison tables make the paper comprehensive
Clarity: Paper is well written and clear. Details make it clear that the environment will be easy for the community to leverage for training and benchmarking.
Significance: With the integration of OpenAI Gym and ability to use this without expertise in Unreal, this work could allow for significant downstream RL benchmarking and explorations that could lead to interesting insights.

**Weaknesses:**

While the types of environments included in UnrealZoo are diverse, there is little diversity in the types of interactions and actions the agents can have in these environments. This limits how complicated these environments can be and how much novelty these environments will drive in terms of RL algorithms explored for learning and becoming experts.

It is a little hard to understand what the "ceiling" for each task is in terms of performance. How hard are the given tasks for current RL algorithms? How long does it take for them to learn and become experts?

Inclusion of results for only DowntownWest feels lacking. Would like to see what performance looks like for other environments too.

**Questions:**

1. How hard are the given tasks for current RL algorithms? With some limited compute, is it possible for RL algorithms to achieve the same performance as a human expert today? These environments can drive downstream RL innovation by just simply being impossible to become experts in today.

2. How long does it take for an RL agent to become an expert in these environments? Another interesting problem to explore in RL is becoming an expert fast. If it takes unreasonably long to become an expert in an environment, then these environments become interesting to share with the community.

---

> ### Author Response · Authors · 2024-11-26
> **Response to Reviewer 7kEB**
>
> Thank you for your valuable feedback. Below, we address each of your points in detail.
>
> > **W1:** There is little diversity in the types of interactions and actions the agents can have in these environments.
>
> **R1:** We appreciate your concern regarding the diversity of interactions and actions in our environments. However, we would like to clarify that our action space already supports a broad range of interactions, such as picking up objects, climbing obstacles, opening and closing doors, entering and exiting vehicles (cars or motorbikes), falling, crouching, PTZ control, and so on. We provide video demonstrations of some typical actions on [this website](https://unrealzoo.notion.site/playable-entities) to demonstrate the built-in actions. These interactions enable us to simulate complex behaviors and tasks in open-world environments, showing its potential to support a wide range of embodied tasks in future research. Furthermore, Unreal Engine's flexibility allows us to customize and add new actions as community needs arise.
>
>
> > **W2&Q1:** It is a little hard to understand what the "ceiling" for each task is in terms of performance. How hard are the given tasks for current RL algorithms?* How hard are the given tasks for current RL algorithms? With some limited computing, is it possible for RL algorithms to achieve the same performance as a human expert today?
>
> **R2:** Thank you for your suggestions. For navigation, we have reported human performance in Table 3 as a reference. There remains a significant performance gap between RL algorithms (A3C) and human performance. For tracking, the ideal ceiling of the episode length and success rate in the current setting should be 500 and 1.00, indicating the tracker continuously following any target in any scene. The episodic reward's upper bound equals the maximum episode length, i.e., 500. At each step, if the tracker accurately places the target in its expected location, the tracker receives a reward of $+1$. Otherwise, the tracker additionally receives a penalty based on position error. We have added the details of the reward function in `Appendix B`. In practice, it is impossible for this reward to reach 500, as the error is inevitable. When the episodic reward reaches 400, it is already a good tracking performance.
>
> To the best of our knowledge, the offline RL solution (Zhong et. al, ECCV 2024) is the best choice to train embodied visual tracking with limited computation resources, requiring only a consumer-level GPU for training. While **current RL algorithms achieve near-optimal performance in simple environments**, the main challenge lies in **generalizing to diverse environments and targets**. The results in Figure 5 demonstrate that RL agents can benefit from increased diversity in training environments and datasets. With the advancements in offline reinforcement learning and the increasing richness of virtual worlds, we believe achieving human-level expert strategies is within reach in the future.
>
> >**W3:** Inclusion of results for only DowntownWest feels lacking. Would like to see what performance looks like for other environments too.
>
> **R3:** For embodied visual tracking, the results in Figure 5 were evaluated in 16 environments (4 types). In the revised appendix, we add the snapshot of the 16 environments (Figure `7` and Figure `8`) and the detailed results ( Table `11`) for more details. For visual navigation, we evaluate the agents in 2 environments (Roof and Factory).  For social tracking, we additionally evaluate the agents in the other 4 unseen environments (Table `12`). In most scenes, tracking performance declines as the number of distractors in the environment increases. The details of these new results and analysis are included in Appendix `D`.
>
> >**Q2:** How long does it take for an RL agent to become an expert in these environments? Another interesting problem to explore in RL is becoming an expert fast. If it takes unreasonably long to be.
>
> **R4:** Thanks for your suggestion. We agree that exploring how to efficiently build an RL agent into an expert is an interesting problem. Thus, our experiments were conducted on a customer-level GPU (GTX 4090), which is affordable to most AI researchers. For embodied visual tracking, it takes 1 hour to collect 100k offline data and 6 hours in training to get the best tracking policy using CQL (a popular offline RL algorithm). For visual navigation, it takes 6 hours to achieve convergence in the **IndustrialArea** environment with four workers using A3C (a popular online RL algorithm). In the **Roof** environment, it takes 3 hours to achieve convergence with six workers. In the Appendix `D.1`, we provide the training curves. Note that applying more advanced GPUs will further reduce the training time.

---

> ### Author Response · Authors · 2024-12-02
>
> We sincerely thank you for your thoughtful review. **As the discussion period draws to a close, we would appreciate your feedback to confirm whether our replies have addressed your concerns.** If you have any remaining questions, we are happy to provide further clarification. If our responses have resolved your concerns, we would be deeply grateful if you could consider raising the score. Thank you again for your time and effort during the review process.

---

### Author Response · Authors · 2024-12-02
**Common Response to All Reviewers**

Dear Reviewers,

We sincerely thank you for your thorough and constructive reviews of our manuscript. We appreciate the time and effort you have invested in providing valuable feedback, which has greatly contributed to improving the quality of our work.

We have carefully considered your comments and have made several revisions to address the concerns raised. Specifically, we have:

- **Enhanced Clarity and Presentation:** We have revised sections of the paper to improve clarity and coherence, ensuring that our contributions and the motivations behind our work are clearly articulated. We have reorganized the appendix and added more details about the environments and experiments. We have also corrected minor issues in the writing and presentation as suggested.  We will conduct a thorough proofreading to enhance the presentation of the next version.

- **Expanded Details of Experiments:** We have included additional qualitative and quantitative results to provide a more comprehensive view of the performance of our agents, including the [demos of agents in various environments](https://unrealzoo.notion.site/task-evt), the [failure case analysis](https://unrealzoo.notion.site/vlm-agent), and more results in Appendix D. This includes clarifications on the [diversity of interactions](https://unrealzoo.notion.site/playable-entities), the implementation details of each agent (Appendix B&C), the time required for training RL agents, and the weakness of current VLM(GPT-4o) agents.

- **Addressed Technical Concerns:** We have provided detailed analyses of simulation speed and the physical capabilities of our platform, emphasizing UnrealZoo's flexibility and potential for exploring emerging topics in embodied AI, including multi-modal reasoning, planning in dynamic environments, and social interactions. The sim2real transferability has already been demonstrated by previous works that also use unreal engines to train agents.

- **Clarified Licensing and Open-Source Strategy**: We have clarified your compliance with licensing agreements and the rationale behind our distribution strategy, which aims to balance customization ability with the need for high-quality content. We will open-source UnrealCV+ and the gym interface we built for the community. We also noted the source of each environment in the [scene gallery](https://unrealzoo.notion.site/scene-gallery) to help potential developers access the source of the content.

We believe these revisions improve our submission and address your concerns. We appreciate your guidance in refining our work and remain committed to contributing to the community.

Thank you once again for your insights. **As the deadline for the open discussion approaches**, we look forward to your feedback on our response. If you have any further concerns, we are open to discussing them with you **before the deadline (Dec 2nd at midnight AoE)**.

Best regards

Authors of UnrealZoo

---

### Meta-Review · Area_Chair_1TMw · 2024-12-24

**Metareview:**

This paper introduces UnrealZoo, a photorealistic and large-scale environment designed for training and evaluating embodied AI agents. Built on the Unreal Engine, the proposed collection includes 100 high-quality 3D scenes and offers diverse playable entities to facilitate embodied AI research. The authors enhance the UnrealCV APIs for better communication efficiency and provide easy-to-use Gym interfaces to accommodate diverse research needs. Experiments conducted on UnrealZoo benchmark tasks such as visual navigation and tracking underscore the value of diverse training environments, while also analyzing the limitations of current RL- and VLM-based agents when working in dynamic and unstructured open worlds.

The reviewers appreciated the variety and realism in UnrealZoo, highlighting its 100 diverse scenes and a range of playable entities such as humans, animals, cars, and robots. They also acknowledged the improvements to the rendering pipeline and enhancements to UnrealCV and agent-environment communication, emphasizing how these could make UnrealZoo a useful tool for embodied AI research.

Reviewers also raised key concerns: the paper misses a clear focus or defined direction, with UnrealZoo not aligning well with sim2real transfer goals or the needs of highly performant RL environments. They mentioned UnrealZoo’s limited practical relevance to real-world deployment platforms and its slow performance. Additionally, binary-only distribution of UnrealZoo was noted as limiting flexibility for third-party modifications. Reviews inquired about missing technical and experimental details. Persistent issues with writing and presentation were also raised. The AC acknowledges that the authors made notable improvements to the writing through multiple rounds of responses.

This submission received scores of 5, 5, 5, and 3 in the initial review phase, and 6, 6, 5, and 3 after the rebuttal phase, with two reviewers (7kEB and u5UD) increasing their ratings. During the post-rebuttal discussion phase, R-u5UD (score 6) noted limited experience in the area, explicitly stated they were "not sure about", and marked their review confidence as 2. R-7kEB (score 6) was asked about potentially championing this paper, but they did not support this. After careful examination and post-rebuttal discussion with reviewers, the AC finds arguments from R-BMxe (score 5) and some from R-dbRo (score 3) most significant for this submission’s evaluation. R-BMxe maintained their "primary concern still stands" regarding the "lack of a clear contribution".

In post-rebuttal discussion, while the AC and the committee did acknowledge writing improvements and some clarifications, they concluded that paper needs more efforts to refine its focus and communicate a clear and compelling contribution to meet ICLR standards.

**Additional Comments On Reviewer Discussion:**

This submission received scores of 5, 5, 5, and 3 in the initial review phase, and 6, 6, 5, and 3 after the rebuttal phase, with two reviewers (7kEB and u5UD) increasing their ratings. During the post-rebuttal discussion phase, R-u5UD (score 6) noted limited experience in the area, explicitly stated they were "not sure about", and marked their review confidence as 2. R-7kEB (score 6) was asked about potentially championing this paper, but they did not support this. After careful examination and post-rebuttal discussion with reviewers, the AC finds arguments from R-BMxe (score 5) and some from R-dbRo (score 3) most significant for this submission’s evaluation. R-BMxe maintained their "primary concern still stands" regarding the "lack of a clear contribution".

---

### Decision · Program_Chairs · 2025-01-22

Reject